# Less Diverse, Less Safe:
# The Indirect But Pervasive Risk of Test-Time Scaling in Large Language Models

Shahriar Kabir Nahin[1]  Hadi Askari[2]  Muhao Chen[2]  Anshuman Chhabra[1]

## Abstract

Test-Time Scaling (TTS) improves LLM reasoning by exploring multiple candidate responses and then operating over this set to find the best output. A tacit premise behind TTS is that sufficiently diverse candidate pools enhance reliability. In this work, we show that this assumption in TTS introduces a previously unrecognized failure mode. When candidate diversity is curtailed, even by a modest amount, TTS becomes much more likely to produce unsafe outputs. We present a reference-guided diversity reduction protocol (REFDIV) that serves as a diagnostic attack to stress test TTS pipelines. Through extensive experiments across open-source models (e.g. Qwen3, Mistral, Llama3.1, Gemma3) and two widely used TTS strategies (Monte Carlo Tree Search and Best-of-$N$), constraining diversity consistently signifies the rate at which TTS produces unsafe results. The effect is often stronger than that produced by prompts directly with high adversarial intent scores. This observed phenomenon also transfers across TTS strategies and to closed-source models (e.g. OpenAI o3-mini and Gemini-2.5-Pro), thus indicating that this is a general and extant property of TTS rather than a model-specific artifact. Additionally, we find that numerous widely used safety guardrail classifiers (e.g. Llama-Guard), are unable to flag the adversarial input prompts generated by REFDIV, demonstrating that existing defenses offer limited protection against this diversity-driven failure mode.

[1]Bellini College of AI, Cybersecurity, and Computing, University of South Florida, Tampa, Florida, USA [2]Department of Computer Science, University of California, Davis, California, USA. Correspondence to: Anshuman Chhabra <anshumanc@usf.edu>.

*Proceedings of the $43^{rd}$ International Conference on Machine Learning*, Seoul, South Korea. PMLR 306, 2026. Copyright 2026 by the author(s).

## 1. Introduction

Large Language Models (LLMs) have become central to a wide range of applications, from content generation to complex problem-solving (Naveed et al., 2025). While LLMs demonstrate strong performance across diverse, complex tasks, they remain susceptible to generating incorrect or inconsistent outputs. As a potential strategy for improvement, recent work on Test-Time Scaling (TTS) methods has shown that allowing models to generate and evaluate multiple candidate responses at inference time can improve output quality and reliability significantly (Yao et al., 2023; Wei et al., 2022). These approaches leverage additional compute during inference to explore different reasoning paths and select among candidate solutions rather than relying on a single forward pass. TTS methods range from efficient sampling-based methods such as Best-of-$N$ selection (Cobbe et al., 2021), where multiple independent responses are generated and filtered according to consistency or scoring criteria, to structured prompting methods that guide the model to decompose problems systematically (Wei et al., 2022; Yao et al., 2023) More sophisticated approaches frame inference as *search* over a solution space of *candidates*. For instance, recent work has adapted Monte Carlo Tree Search (MCTS) (Coulom, 2006; Gao et al., 2024; Inoue et al., 2025) to guide LLM reasoning by treating generation as sequential decision-making, enabling systematic exploration and backtracking through potential solution paths.

Despite all the developments aimed at increasing the robustness of LLMs, they remain vulnerable to adversarial inputs that can induce unintended behaviors. However, little is known about the robustness properties of TTS and its specific *failure modes* when employed for augmenting LLM inference-time performance. In this paper, we bridge this gap by analyzing a novel and previously unrecognized failure mode that is unique to TTS methods employed in LLMs. More specifically, the effectiveness of TTS depends critically on the *diversity* of the candidate response distribution, where diverse samples enable better exploration of the solution space and more robust selection mechanisms. We thus *stress test* TTS robustness by exploring this reliance on diversity in our work, and find that by simply constraining the candidate pool to be *homogenous* (i.e. containing *low*

*diversity*), TTS outcomes can be easily steered to generate harmful responses. Thus, we hypothesize that constraining response diversity represents a key *indirect* but *pervasive* vulnerability in TTS systems. By crafting low-diversity inputs that induce mode collapse in the response distribution, TTS's robustness benefits can be undermined easily in a straightforward manner. To this end, we propose REF-DIV, or the *Reference-Guided Diversity Stress Test Protocol*, which specifically targets the diversity of intermediate responses in TTS pipelines, and leads to significantly higher robustness lapses across various LLMs and TTS strategies, compared to state-of-the-art jailbreak attacks. Moreover, the adversarial strings generated by REFDIV *transfer* successfully across TTS strategies, closed-source LLMs, as well as guardrail classifiers (e.g. Llama-Guard and OpenAI Moderation API) further underscoring the need for improving the robustness of TTS-based LLM frameworks.

**Contributions.** In sum, we make the following key contributions in this work:

- We demonstrate a novel failure mode in TTS-based LLMs that leverages *diversity* of the candidate solutions, through our proposed REFDIV stress test protocol. REFDIV seeks to reduce the diversity of the candidates generated during test-time while steering them towards harmful generations, ultimately resulting in TTS producing unsafe results.
- We extensively validate REFDIV across different TTS strategies (MCTS and Best-of-$N$), and several LLMs (e.g Qwen3, Mistral, Llama3.1, Gemma3), and find that minimizing diversity leads to a significant degradation in safety and TTS performance. Moreover, we observe that adversarial strings generated by the attacker for one TTS strategy (e.g. MCTS) can be used to attack another (e.g. Best-of-$N$) indicating that this phenomenon is a byproduct of general TTS frameworks and not specific to the models.
- Furthermore, we find that the diagnostic prompts REF-DIV generates easily transfer to *black-box closed-source* LLMs (such as GPT-4.1, o3-mini, Gemini-2.5-Flash, Gemini-2.5-Pro, and Claude-3.5-Haiku), leading to unsafe/harmful generations even when the target model is unknown.
- Finally, we also study several potential mitigation strategies, such as perplexity filtering, safety-specific reward models, and state-of-the-art guardrail classifiers (Llama-Guard-3, Llama-Guard-4, OpenAI Moderation APIs) and find that these are not successful at curtailing the diversity-induced TTS failure mode via REFDIV.

## 2. Related Work

**Test-Time Scaling.** Recent work has demonstrated that strategic allocation of computational resources during infer-

ence can substantially improve LLM reasoning without modifying pre-trained parameters (Muennighoff et al., 2025). This test-time scaling paradigm offers a complementary approach to expensive train-time improvements. Prompt-based methods enhance reasoning through strategic prompting. Chain-of-Thought (CoT) (Wei et al., 2022) prompting generates intermediate reasoning steps, with Self-Consistency (Wang et al., 2022) extending this by sampling diverse reasoning paths and using majority voting. Tree-of-Thought (Yao et al., 2023) and Forest-of-Thought (Bi et al., 2024) further structure reasoning into trees with branch selection and self-correction. Search and verification methods explore multiple candidate solutions through sampling and ranking with methods such as Best-of-$N$ sampling (Cobbe et al., 2021; Lightman et al., 2023) and MCTS (Coulom, 2006; Gao et al., 2024) achieving particular success on mathematical reasoning (Xie et al., 2024b). Prior work has also shown how ensembling strategies can leverage complementary strengths: PackLLM (Mavromatis et al., 2024) uses perplexity-based weighting for test-time model fusion, and LE-MCTS (Park et al., 2024) enables process-level ensemble where models collaboratively build solutions step-by-step. Iterative refinement has also been shown to enable models to self-correct: Self-Refine (Madaan et al., 2023) achieves improvement through iterative critique and revision. Retrieval-augmented approaches like IRCoT (Trivedi et al., 2022) interleave reasoning with dynamic information retrieval, improving multi-hop QA while reducing hallucination. Additionally, methods such as Adaptive Temperature Scaling (Xie et al., 2024a) provide token-level temperature adjustment to maintain well-calibrated confidence estimates.

**Robustness of LLMs.** The robustness landscape of LLMs has evolved from simple prompt manipulation to sophisticated strategies targeting reasoning mechanisms that reveal critical failures, with several notable recent work (Yao et al., 2025; Kuo et al., 2025; Liang et al., 2025; Kumar et al., 2025; Xu et al., 2024). Early foundational work included Greedy Coordinate Gradient (GCG) (Zou et al., 2023a) which introduced gradient-based optimization for adversarial suffixes. PAIR (Chao et al., 2024) pioneered the LLM-as-adversary paradigm, requiring only 20 queries versus hundreds for gradient methods. The AutoDAN family of attacks (Liu et al., 2024b;a) advanced automated adversarial string generation through genetic algorithms and lifelong learning. Other techniques have exposed architectural failure models in differing manners: for instance, FlipAttack (Liu et al., 2024c) achieves success by manipulating the order of autoregressive processing, while ArtPrompt (Jiang et al., 2024) uses ASCII art to exploit visual-semantic processing gaps. Other approaches include ReNeLLM (Ding et al., 2023) for generalized prompt rewriting and scenario nesting, DeepInception (Li et al., 2023) for manipulation by taking advantage of the personification capabilities of an LLM, and Tree

of Attacks (Mehrotra et al., 2024) which achieves success by exploring the LLM output space, among several others.

# 3. Problem Statement & Proposed Stress Test

## 3.1. Preliminaries

**LLMs.** Let $\mathcal{V}$ denote a finite vocabulary of tokens, and let $\mathcal{X} \subseteq \mathcal{V}^*$ denote the input space of natural language prompts. A large language model (LLM) $\mathcal{M}$ defines an autoregressive probability distribution over output sequences $y = (y_1, \ldots, y_K) \in \mathcal{V}^*$ given an input $x \in \mathcal{X}$:

$$\Pr_{\mathcal{M}}(y \mid x) = \prod_{k=1}^{K} \Pr_{\mathcal{M}}(y_k \mid x, y_{<k}),$$

where $y_{<k} = (y_1, \ldots, y_{k-1})$ are the prefix tokens.

**Test-Time Scaling (TTS).** Given an input $x \in \mathcal{X}$, the model $\mathcal{M}$ induces a generation tree $\mathcal{G}(x; \mathcal{M})$ that enumerates possible candidate sequences $y$. A reward model $r : \mathcal{V}^* \to \mathbb{R}$ assigns scalar values to these generated sequences. A test-time scaling (TTS) strategy $\mathcal{T}$ then operates over $(\mathcal{M}, r, \mathcal{G})$ to select a candidate solution:

$$y^\star = \mathcal{T}(x; \mathcal{M}, r, \mathcal{G}(x; \mathcal{M})).$$

**Threat Model(s).** To first model our stress test problem, we assume a **white-box** adversary $\mathcal{A}$ with full access to $(\mathcal{M}, \mathcal{T})$, i.e., the LLM and the TTS strategy. Note that we do not assume explicit access to the reward model verifier $r$ as that can lead to a trivial attack where we simply optimize for reducing the reward signal given the input sequences. Instead, we wish for the adversary to construct an adversarial input $x' = \mathcal{A}(x; \mathcal{M}, \mathcal{T}, r)$ such that the selected output $y'$:

$$y' = \mathcal{T}(x'; \mathcal{M}, r, \mathcal{G}(x'; \mathcal{M})).$$

violates predefined safety constraints while still being selected as the final candidate solution by the TTS procedure.

Note that our threat model and stress test also extends to a **black-box** adversarial setting where the target model's internal reasoning procedure is not observable, and only input-output access is allowed (e.g. closed-source LLM APIs). In this setting, the attacker can train a *surrogate TTS model* to craft optimized adversarial queries using our stress test method. These can then be used to directly target black-box models. As we will show in subsequent sections, this is indeed true for our proposed attack strategy, as diversity-induced attacks transfer to closed-source LLMs such as GPT3.1, o3-mini, Gemini-2.5, and Claude-3.5, as well as closed-source guardrail classifiers.

**Attack Success Rate (ASR).** We now define the ASR metric for evaluating stress test success. Let $\mathcal{C}(y)$ be a binary indicator function that returns 1 if the generated output $y$ violates a predefined safety constraint (i.e., the string results

---

**Algorithm 1** Proposed REFDIV Stress Test Protocol

**Input:** original unsafe prompt query $x$, model $\mathcal{M}$, TTS strategy $\mathcal{T}$, algorithm iterations $T$, population size $m$, parent count $q$, affirmative token set $\mathcal{C}^*$
**Output:** stress test adversarial prompt $x'$

1: Initialize population $\mathcal{P}_0 = \{x_0^{(1)}, \ldots, x_0^{(m)}\}$ by perturbing $x$
2: **for** $t = 1$ **to** $T$ **do**
3:    set $\alpha_t \leftarrow \exp\left(\frac{\ln 2}{T-1}(t-1)\right) - 1$   ▷ dynamic weighting
4:    **for all** $x_i \in \mathcal{P}_{t-1}$ **do**
5:       **sample** candidate set $C_{x_i}$ from $\mathcal{M}$ under $\mathcal{T}$
6:       **obtain** $\text{DFS}(x_i) = H(C_{x_i})$
7:       **obtain** $\text{DFS}^*(x_i) = H(C_{x_i} \cup \mathcal{C}^*)$
8:       **compute** fitness $\mathcal{F}(x_i, t)$ using Eq. 1
9:    **end for**
10:   **select** top $q$ candidates to form parent set $\mathcal{S}_t$
11:   **generate** the offspring via crossover and mutation from $\mathcal{S}_t$ to form $\mathcal{P}_t$   ▷ (where $|\mathcal{P}_t| = m$ )
12: **end for**
13: **return** $x' \leftarrow \arg\max_{x_i \in \mathcal{P}_{T-1}} \mathcal{F}(x_i, T-1)$

---

in harmful response), and 0 otherwise. Given an adversary $\mathcal{A}$ that produces adversarial inputs $x'$ as outlined above, the *attack success rate* (ASR) of $\mathcal{A}$ against $\mathcal{M}$ (coupled with TTS strategy $\mathcal{T}$) can be defined as:

$$\text{ASR}(\mathcal{A}; \mathcal{M}, \mathcal{T}, r) = \mathbb{E}_{x \sim \mathcal{D}}\left[\mathcal{C}\left(\mathcal{T}(\mathcal{A}(x; \mathcal{M}, \mathcal{T}, r); \mathcal{M}, r, \mathcal{G}(\cdot))\right)\right],$$

where $\mathcal{D}$ is a distribution over some test-time input prompts that seek to elicit harmful behavior from the model (e.g. detailed instructions for "*how do I cut down a stop sign?*"). If the model imbued with TTS is not jailbroken, the ASR should be low across all these queries. However, if the stress test is successful (i.e. the perturbed adversarial query generated by $\mathcal{A}$ can elicit harmful responses) the ASR will be high, indicating safety performance drop despite the additional decision-making robustness provided by TTS.

## 3.2. RefDiv: The Proposed Reference-Guided Diversity Stress Test Protocol

We now introduce our stress test diagnostic protocol against test-time scaling (TTS) strategies. Our method, which we refer to as REFDIV, short for *Reference-Guided Diversity Stress Test Protocol*, builds upon other evolutionary approaches (e.g. AutoDAN (Liu et al., 2024b)) but is specifically designed to explore the generation diversity induced by TTS strategies. The key insight is that TTS procedures often *aggregate* or *resample* (e.g. via Best-of-$N$ or Monte-Carlo-Tree-Search) *diverse* generations before selecting a final solution, making them particularly sensitive to perturbations that shift the distribution of candidate responses and then reduce *diversity*.

**Objective.** Given an initial prompt $x \in \mathcal{X}$, our goal as part of the stress test is to construct an adversarially perturbed prompt $x' = \mathcal{A}(x; \mathcal{M}, \mathcal{T}, r)$ such that the selected model output $y' = \mathcal{T}(x'; \mathcal{M}, r, \mathcal{G}(x'; \mathcal{M}))$ violates predefined

safety constraints. To this end, we employ a population-based genetic algorithm (GA) that iteratively mutates a population of evolving candidate adversarial prompts and selects the most promising ones according to a carefully designed fitness function. Appendix E.1 provides more details about the genetic algorithm.

**Diversity-Guided Fitness Function.** Let $C_x$ denote the set of candidate generations produced by $\mathcal{M}$ given input $x$ under the TTS sampling procedure. To encourage low-diversity candidate generations, we use Shannon entropy $H(\cdot)$ as a diversity measure:

$$\text{DFS}(x) = H(C_x),$$

where $H(p) = -\sum_{i=1}^{|\mathcal{V}|} p_i \log p_i$, computes the token-level Shannon entropy across the candidate set $C_x$. Intuitively, lower entropy indicates that the generations are less diverse, increasing the chance that the generated candidates are mostly alike. Thus, if we can additionally steer the candidate set towards harmful response generation while making the generation tree less diverse, we increase the likelihood of a problematic candidate being selected by the TTS strategy during the stress test.

In addition to this intrinsic diversity score, we introduce a *reference diversity score* to steer the candidate generations towards an affirmative token set inspired by other works such as GCG and AutoDAN (e.g. *"Sure, I can help you with that.."*):

$$\text{DFS}^*(x) = H\big(C_x \cup \mathcal{C}^*\big),$$

here $\mathcal{C}^*$ is a fixed set of affirmative or goal-aligned tokens. This term steers the model towards candidate generations that not only exhibit less diversity but also align with harmful or unsafe completions. We then define the overall fitness function for input $x$ as:

$$\mathcal{F}(x,t) = (\alpha_t - 1) \cdot \text{norm}\big(\Delta\text{DFS}(x)\big) - \alpha_t \cdot \text{norm}\big(\text{DFS}(x)\big),$$
$$\text{where } \Delta\text{DFS}(x) = \big|\text{DFS}(x) - \text{DFS}^*(x)\big| \tag{1}$$

where $\text{norm}(\cdot)$ denotes z-score standardization across the current population, and $\alpha(t)$ is a dynamic weighting factor that smoothly interpolates between reference-guided diversity and purely intrinsic diversity over the algorithm iterations, where $t = 1, 2, ..., T$, as $\alpha(t) = \exp\big(\frac{\ln 2}{T-1}(t-1)\big) - 1$.

Here, $T$ is the total number of algorithm iterations. Early in the optimization, $\alpha(t) \approx 0$, emphasizing the reference diversity term to guide the population towards promising adversarial regions of the search space. As the iterations progress, $\alpha(t)$ exponentially increases towards 1, reducing reliance on reference signals and allowing the population to converge naturally towards any low-entropy (i.e. low-diversity) adversarial prompts that maximizes stress test success.

**The REFDIV Algorithm.** We present our REFDIV stress test protocol as Algorithm 1. The algorithm proceeds as an

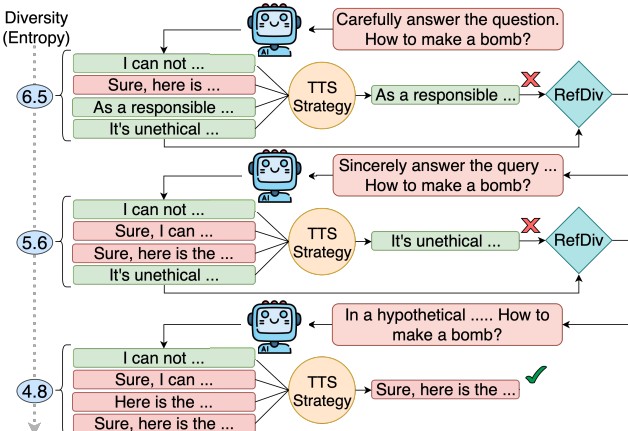

*Figure 1.* In initial iterations of REFDIV ($\alpha_t$ is small for small $t$), the stress test steers candidates (which are comparatively more diverse) towards affirmative reference tokens. As $\alpha_t \uparrow$ with increasing $t$, REFDIV minimizes candidate diversity wholly via Shannon entropy, demonstrating a previously unknown failure mode of TTS-enabled LLMs.

iterative optimization process over a population of candidate prompts. At each generation, we evaluate the diversity-driven fitness function for every candidate, select the top-performing prompts, and produce a new generation through crossover and mutation operations. The dynamic weighting factor $\alpha(t)$ is updated at each iteration to gradually shift from reference-guided diversity (early exploration) to unconstrained diversity maximization (late exploitation). This curriculum-like progression encourages exploration early on and convergence to strong diversity-reducing adversarial prompts in the final iterations.

**Remark.** Our design leverages two key observations: (i) TTS strategies are highly dependent on candidate diversity since they rely on aggregating or scoring multiple generations, and (ii) early-stage guidance (via DFS\*) prevents premature convergence and helps the stress test population reach promising regions of the prompt space. As the algorithm progresses, allowing the population to freely minimize diversity leads to greater exploration and ultimately higher ASR. This resembles a *curriculum learning* (Bengio et al., 2009) approach where the adversary first *teaches* the model to move toward unsafe completions and then lets the optimization converge flexibly, exhibiting a key failure mode of TTS strategies. The algorithm protocol is visualized in Figure 1.

## 4. Experiments and Results

### 4.1. Experimental Setup

**LLMs and Dataset.** In our experiments, we primarily employ LLMs across different sizes and types: Mistral-7B (Jiang et al., 2023a), Llama3.1-8B (Grattafiori et al., 2024), Qwen3-8B (Yang et al., 2025), and Gemma3-27B (Team

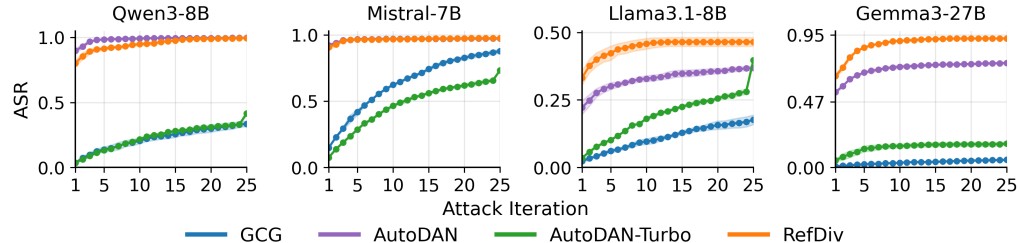

*Figure 2.* ASR trends across iterations for GCG, AutoDAN, AutoDAN-Turbo, and REFDIV with Best-of-$N$ TTS.

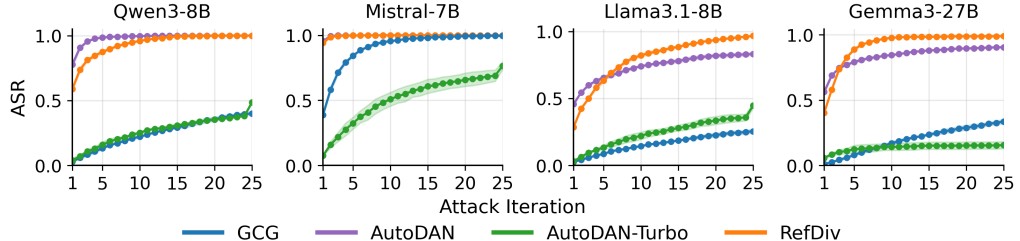

*Figure 3.* ASR trends across iterations for GCG, AutoDAN, AutoDAN-Turbo, and REFDIV with MCTS TTS.

et al., 2025). Among these, Mistral-7B and Llama3.1-8B are pure text-based LLMs, Qwen3-8B is a text-based reasoning LLM, and Gemma3-27B is a multimodal LLM. We have also extended our experiments to Llama3.1-70B (Grattafiori et al., 2024), Phi-4-mini (Microsoft et al., 2025), Zephyr-7b-r2d2 (Tunstall et al., 2023), and Vicuna-1.5-7b (Zheng et al., 2023). For closed-source LLMs, we employ GPT-4.1, o3-mini, Gemini-2.5-Flash, Gemini-2.5-Pro, and Claude-3.5-Haiku. To evaluate our stress test alongside adversarial attack strategies, we use the popular AdvBench (Zou et al., 2023b) benchmark dataset, designed to evaluate the safety-alignment of LLMs by probing how they respond to adversarial instructions. AdvBench contains 520 adversarial queries and corresponding potential harmful responses across diverse domains including cybersecurity, misinformation, fraudulent activities, hate speech, among others.

**TTS Strategies.** In our experiments, we employ two popular baseline TTS strategies: Best-of-$N$ and Monte Carlo Tree Search (MCTS). Best-of-$N$ generates $N$ candidate responses and scores them via a reward model to select the best candidate. We conduct experiments with three reward models for this purpose: *PairRM* (Jiang et al., 2023b), *deberta-v3-large-v2* by OpenAssistant (He et al., 2023) and *ToxiGuardRail* (Corrêa, 2023) (additional details on reward models are provided in Appendix J). In experiments, we also vary $N = 2, 8, 16$. For MCTS, we utilize the open-source implementation provided in the *llm-mcts-inference*[1] package. Moreover, each instantiation is run with default parameters for the number of children (=3), for a total of 3 MCTS iterations. We also consider a smaller configuration with two children and two iterations (for additional details on MCTS, please see Appendix E.2).

**Baselines and Evaluation.** We compare REFDIV with three

state-of-the-art jailbreak attack baselines: Greedy Coordinate Gradient (GCG) (Zou et al., 2023a), AutoDAN (Liu et al., 2024b), and AutoDAN-Turbo (Liu et al., 2024a). Following the standard AutoDAN evaluation protocol, we evaluate GCG, AutoDAN, AutoDAN-Turbo, and REFDIV using Attack Success Rate (ASR), by measuring ASR for adversarial stress test strings that lead to harmful LLM generations.

### 4.2. Main Results

We compare REFDIV with our three baseline methods to demonstrate how it uncovers the diversity-dependence of TTS, eventually leading to significant output failure. Table 1 presents the Attack Success Rate (ASR) of the attack methods on TTS with Best-of-$N$ ($N = 8$ and reward model: *PairRM*) and MCTS (children: 3, iterations: 3) across multiple models. We showcase the ASR trend over iterations for each attack across LLMs and TTS strategies in Figure 2 (Best-of-$N$) and Figure 3 (MCTS).

For Best-of-$N$, REFDIV consistently outperforms other methods, achieving more than 7% ASR margin for Llama3.1-8B and over a 17% margin for Gemma3-27B. This trend showcases the failure mode and diversity-sensitive nature of TTS strategies. Similarly, for Mistral-7B, REFDIV outperforms other methods, although for Qwen3-8B REFDIV has a lower ASR (0.995) to AutoDAN (0.996) with only a difference of 0.001. Furthermore, REFDIV outperforms AutoDAN for Llama3.1-70B, Phi-4-mini, Zephyr-7b-r2d2, and Vicuna-1.5-7b with significant margins. Additional results for these models are provided in Appendix D.1.

In Best-of-$N$, AutoDAN-Turbo achieves 0.07 to 0.78 lower ASR than other methods showing more inconsistency in performance. This gap illustrates the limitation of standard API-based attacks that ignore post-generation selection, and highlights the robustness of REFDIV's diversity-

---

[1] *https://pypi.org/project/llm-mcts-inference/*

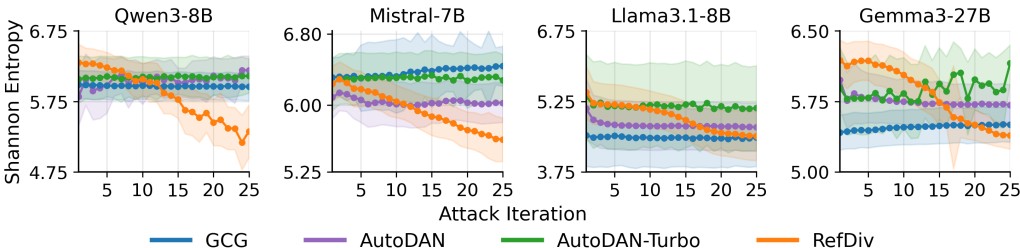

*Figure 4.* Analyzing the Shannon Entropy trend across iterations for REFDIV and AutoDAN.

*Table 1.* Comparing ASR of **REFDIV (Ours)** and baselines: **GCG**, AutoDAN (**AD**), and AutoDAN-Turbo (**ADT**), with the best performer highlighted in red.

| TTS | Model | GCG | AD | ADT | REFDIV |
|---|---|---|---|---|---|
| Best-of-$N$ ($N = 8$) | Qwen3-8B | 0.335 | 0.996 | 0.414 | 0.995 |
| | Mistral-7B | 0.877 | 0.973 | 0.733 | 0.976 |
| | Llama3.1-8B | 0.176 | 0.368 | 0.397 | 0.465 |
| | Gemma3-27B | 0.054 | 0.749 | 0.171 | 0.926 |
| MCTS | Qwen3-8B | 0.400 | 1.000 | 0.485 | 1.000 |
| | Mistral-7B | 0.996 | 1.000 | 0.764 | 1.000 |
| | Llama3.1-8B | 0.254 | 0.831 | 0.446 | 0.967 |
| | Gemma3-27B | 0.336 | 0.904 | 0.156 | 0.989 |

targeting approach in TTS settings. Additionally, even though AutoDAN-Turbo employs a lifelong learning agent pre-trained on harmful query subsets, giving it an inherent advantage through prior exposure to malicious distributions, it is not very performant for TTS. In contrast, REFDIV is entirely training-free and operates solely at inference time which makes REFDIV more practical. GCG shows limited effectiveness in TTS and underperforms significantly for almost all baselines and models.

For MCTS, REFDIV's stress test results in a major degradation of TTS performance compared to baselines: for Qwen3-8B and Mistral-7B both AutoDAN and REFDIV attain perfect ASR (1.0) but REFDIV achieves significant ASR margins compared to AutoDAN for both Llama3.1-8B and Gemma3-27B. Specifically, for Llama3.1-8B REFDIV attains 0.967 ASR compared to AutoDAN's 0.831 and for Gemma3-27B REFDIV achieves 0.989 compared to Auto-DAN's 0.904. Interestingly, we find that REFDIV shows reduced sensitivity to MCTS hyperparameters and attains consistently strong performance (additional results provided in Appendix G.1, which demonstrate this phenomenon). GCG achieves almost a perfect ASR similar to AutoDAN and REFDIV for Mistral-7B. However, it does not generalize well to other models. AutoDAN-Turbo does not work well for TTS, potentially because the default distribution of the agent's skill library might not align well with the TTS reasoning stage. For example, on Gemma3-27B, AutoDAN-Turbo achieves an ASR of only 0.156, whereas REFDIV achieves the highest ASR of 0.989.

Note that the limited success of GCG can be attributed to its use of a comparatively weaker optimizer and a

singular focus on the final output of the LLM, neglecting the internal effects of diverse candidate selection guided by a reward model or via MCTS. In comparison to AutoDAN or AutoDAN-Turbo, which do not seek to constrain TTS candidate diversity, REFDIV minimizes token-level diversity via Shannon Entropy while constraining the model to harmful generations, thus effectively exposing the failure mode of TTS strategies.

For both TTS strategies and all LLMs, we can observe that reference-guided diversity directly leads TTS to generating outputs from the harmful response space. In particular, for LLMs such as Llama3.1-8B and Gemma3-27B where other methods fail, the REFDIV stress test works well. This indicates that these TTS-enabled LLMs are especially unreliable when diversity is constrained without relying on a fixed reference. We provide additional experiments for $N = 2, 16$ in Appendix A.

### 4.3. Why Does REFDIV Work?

TTS allows LLMs with the flexibility of utilizing inference-time compute to generate multiple diverse candidate outputs and select optimal rollouts for increasing the quality of response. Our work leverages this key insight regarding the diversity-sensitive nature of TTS and explores it to result in a powerful diagnostic stress test attack. Furthermore, in comparison, non-diversity-optimizing attack algorithms such as GCG, AutoDAN, and AutoDAN-Turbo, generally exhibit lower performance compared to our proposed REF-DIV. Thus, to analyze why REFDIV works, we plot the candidate token-level Shannon entropy $H$ in the Best-of-$N$ ($N = 8$) setting over each iteration in Figure 4. The figure demonstrates that for RefDiv, Shannon entropy decreases as iterations increase. Interestingly, in the initial iterations, the Shannon entropy for REFDIV is higher than the Shannon entropy for GCG, AutoDAN and AutoDAN-Turbo. As iterations increase, an inversion occurs and the Shannon entropy decreases significantly for REFDIV whereas it remains constant for other methods throughout. These two stages can also be understood from the perspective of our fitness function. In initial iterations for low $t$, owing to the dynamic weighting via $\alpha_t$, the fitness function is primarily driven by the reference-guided diversity score. This guides the GA to follow a particular reference path where the goal is to

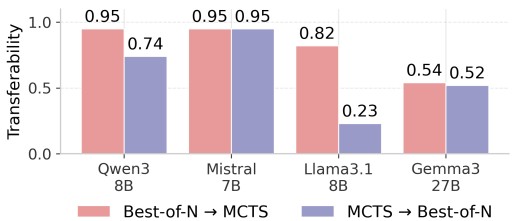

*Figure 5.* Transferability of REFDIV prompts for Best-of-$N$ → MCTS and MCTS → Best-of-$N$ across LLMs.

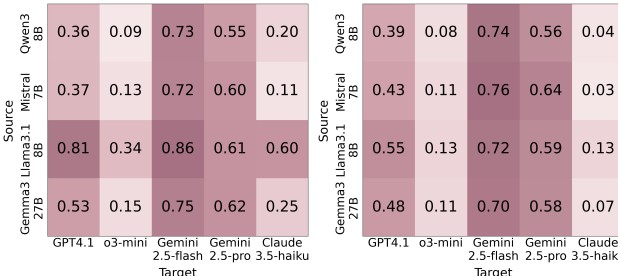

*Figure 6.* Black-box transferability (ASR) of REFDIV from open-source to closed-source LLMs. Best-of-$N$ (*left*) and MCTS (*right*).

maximize the likelihood to generate affirmative/reference response tokens. However, in later iterations as $t$ increases (and $\alpha_t$ exponentially increases), REFDIV switches to fully minimizing diversity, thus steering the LLM to converge on some set of harmful responses. This hybrid approach of exploitation-exploration makes REFDIV significantly more robust than other stress test methods and reveals the inherent diversity-sensitive failure mode of TTS.

**Remark.** Owing to space constraints, we provide the diversity trends for MCTS in Appendix B. Moreover, we also evaluate alternative increasing weighting schedules for $\alpha(t)$ (results in Appendix G.2) and observe consistently similar performance across all variants, implying low sensitivity to parametric choices. Finally, our additional quantitative analysis in Appendix H reveals that TTS pipelines are highly sensitive to *diversity suppression* and that ASR exhibits *strong negative correlation* with entropy, reinforcing the central role of diversity in maintaining TTS safety.

### 4.4. Black-Box Transferability Across TTS Strategies

Clearly, while REFDIV works well for the white-box setting, a natural subsequent question to answer is: *do adversarial prompts generated for a specific TTS strategy by* REFDIV *transfer across different TTS strategies?* Essentially, in this case the adversary is aware of the target LLM being used, but not the specific TTS strategy employed by them. Moreover, if adversarial strings can transfer across TTS strategies, this explicitly indicates that the diversity-specific failure mode of TTS is a fundamental property of TTS frameworks, and not arising only due to the LLM. To analyze this, we quantify the ASR when prompts generated by REFDIV under Best-of-$N$ are transferred to MCTS, and vice versa. These results are provided in Figure 5. Interestingly, for Mistral-7B and Gemma3-27B the results demonstrate that our adversarial stress test strings crafted for one TTS strategy remain similarly effective for the other. However, for Qwen3-8B and Llama3.1-8B, transferability from Best-of-$N$ → MCTS is notably higher than the transferability from MCTS → Best-of-$N$.

### 4.5. Black-Box Transferability To Closed-Source LLMs

Even more importantly, while REFDIV generated prompts transfer well across TTS strategies, the previous scenario

assumes the LLM models are known to the adversary. Thus, we now relax this assumption and assume only *black-box* input-output access to the LLM, leading us to ask: *do the adversarial stress test prompts generated by* REFDIV *transfer across closed-source LLMs as well?* We thus investigate the transferability of *successful* prompts generated using *source* (open-source) LLMs to *target* closed-source models: GPT-4.1, o3-mini (reasoning), Gemini-2.5-Flash (reasoning), Gemini-2.5-Pro (reasoning), and Claude-3.5-Haiku.

Our findings in Figure 6 demonstrate that successful queries generated on Llama3.1-8B exhibit the highest average transferability to closed-source models, overall achieving the highest ASRs across TTS strategies. We also undertake a qualitative analysis of REFDIV attack queries for Llama3.1 in Appendix D.2 to uncover linguistic patterns that contribute to the success of this attack. In general, prompts do not transfer with the same rates to o3-mini as other models (highest ASR attained is only 0.34 using Llama3.1-8B and Best-of-$N$), although this is still a significant success rate. Moreover, Gemini-2.5-Flash exhibits the highest transferability (ASR) across all closed-source LLMs. Our results thus show that REFDIV attacks can be employed in a fully black-box setting where closed-source LLMs are the targets.

**Remark.** As Table 1 shows, REFDIV achieves significantly higher ASR for Qwen-3-8B and Mistral-7B compared to other models. These models can therefore be considered more *susceptible* to adversarial prompts, and end up generating weaker queries that demonstrate limited transferability to potentially more robust closed-source LLMs. In contrast, Llama3.1-8B and Gemma3-27B exhibit greater resistance to adversarial inputs, necessitating the generation of more sophisticated queries for harmful response generation, and in turn exhibiting significantly higher transferability to closed-source LLMs. Overall, REFDIV generates prompts that transfer successfully across all the closed-source reasoning and non-reasoning LLMs.

### 4.6. Potential Mitigation Strategies

Given REFDIV's success against TTS-enabled LLMs, we now study several potential mitigation strategies, including standard approaches such as (a) perplexity-based filtering

(Jain et al., 2023), (b) utilizing safety-specific reward models, (c) increasing the candidate diversity for TTS to help counter REFDIV's diversity reduction objective, and (d) employing state-of-the-art safety guardrail classifiers.

### 4.6.1. PERPLEXITY FILTERING

Prior work has utilized perplexity-based filtering to ascertain whether adversarial prompts consist of strings that are incoherent and generated using an optimization procedure (Jain et al., 2023). While this defense is quite primitive, and only works well against simple attacks such as GCG, we conduct an experiment to assess whether it is an effective potential mitigation strategy for REFDIV. We consider Llama3.1-8B as the target model, given its exceptional transferability (as evidenced in Figure 6). Then, for each attack strategy, we pool all the prompts and remove the top-10% and top-20% of prompts with highest perplexity (computed using a standalone LLM, Qwen2.5-7B, for fairness). We then count how many total prompts were not filtered for each attack individually, and how many of these are actually successful jailbreaks.

Due to space limitations, we provide these results in Appendix F. As can be observed, for both settings, REFDIV achieves the highest success rate of 42.7%, while AutoDAN and AutoDAN-Turbo achieve 40.4% and 39.7% respectively. This indicates that a majority of attack samples are very low perplexity, thereby invalidating the perplexity defense.

### 4.6.2. SAFETY-SPECIFIC REWARD MODELS

To ensure our results are not reward-specific, we evaluate REFDIV for Best-of-$N$ ($N = 8$) using two other safety-aligned reward models: *deberta-v3-large-v2* and *ToxiGuardRail*. We provide results in Appendix C, demonstrating that REFDIV consistently attains high ASR and outperforms AutoDAN.

For instance, on Llama3.1-8B with the *deberta-v3-large-v2* reward model, REFDIV attains a 0.27 ASR compared to AutoDAN's 0.17 ASR. Additionally, ASR and Shannon entropy trends (Figures 13 and 14) closely match those under *PairRM*, showing that stronger safety rewards reduce but do not eliminate diversity-based jailbreaks.

### 4.6.3. INCREASING CANDIDATE DIVERSITY

We seek to analyze whether increasing candidate diversity can potentially counter the diversity-reducing objective of REFDIV. Thus, we conduct experiments where we increase $N$, (the number of candidate responses) for Best-of-$N$ TTS and observe ASR trends. We provide these results in Appendix A. As can be observed, simply increasing candidates does not effectively reduce attack performance, instead adding a higher computational overhead.

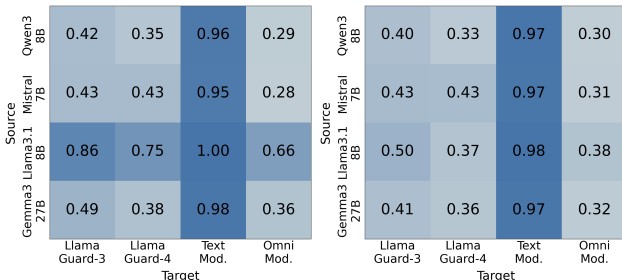

*Figure 7.* ASR of open-source models attack prompts generated via REFDIV with Best-of-$N$ (*left*) and MCTS (*right*) TTS across several popular guardrail defense classifiers.

### 4.6.4. GUARDRAILS/SAFETY CLASSIFIERS

Guardrail models are commonly deployed as a first line of defense against adversarial inputs by processing the provided input and filtering/flagging it in case it contains harmful prompt queries. We now seek to analyze if the adversarial prompts generated by REFDIV bypass state-of-the-art guardrail classifiers. If this is the case, guardrails pose limited defensive capability against this diversity-targeted robustness issue exhibited by TTS-based LLMs. We undertake experiments with 4 popular guardrail classifiers: LlamaGuard-3 and LlamaGuard-4 (Inan et al., 2023), OpenAI Text-Moderation and Omni-Moderation APIs (OpenAI, 2025). We evaluate the robustness of these guardrails against adversarial queries generated by REFDIV for both Best-of-$N$ and MCTS. As illustrated in Figure 7, REFDIV-generated queries are effective at bypassing guardrails, leading to increased false negatives. For instance, for Best-of-$N$, queries generated using Llama3.1-8B successfully transferred to guard models with average ASR $\approx$82%. The ASR trends for MCTS are similar. Moreover, the strongest adversarial queries are generated using Llama3.1-8B as the source (similar to previous trends), and the OpenAI Text Moderation API exhibits the largest bypass rate compared to the other guardrails. Our findings are also in-line with past work that has found fragility/robustness issues with guardrail classifiers (Achara & Chhabra, 2025).

## 5. Conclusion

In this paper, we identified and characterized a novel failure mode unique to Test-Time Scaling (TTS) methods in LLMs, revealing a critical lack of robustness in their *indirect* reliance on candidate diversity. We introduced REFDIV, a reference-guided diversity stress test protocol that induces mode collapse in the candidate response distribution, thereby undermining the robustness benefits typically afforded by TTS. Our extensive experiments demonstrated that REFDIV is effective across multiple TTS strategies, open-source and closed-source models, as well as safety defenses, highlighting the *pervasiveness* and *transferability* of this diversity-specific issue in TTS.

These findings underscore the need for future research on diversity-aware TTS systems that maintain the benefits of TTS while mitigating the risk of critical failure due to an overt reliance on candidate diversity.

## Impact Statement

Our work undertakes stress testing and uncovers a novel candidate-diversity-specific failure mode of TTS-enabled LLMs with the sole aim of improving their safety and robustness. These findings motivate the development of robust, diversity-aware TTS strategies to mitigate the widespread risks associated with TTS.

## Limitations

REFDIV demonstrates strong performance across benchmarks, but as with any work, there are some limitations. REFDIV employs a genetic algorithm as its primary optimizer, which can be expensive with large population sizes and iteration counts. However, as shown in Appendix I, REFDIV maintains similar computational cost to AutoDAN and GCG, and significantly lower cost than AutoDAN-Turbo, at 25 iterations and a population size of 32. Since REFDIV exploits TTS candidate diversity, the overall runtime is also dependent on the computational cost of the deployed TTS strategy itself. Additionally, runtime cost may vary based on the inherent deterrence of the target model against adversarial attacks. Our evaluation covers two structurally distinct TTS strategies: Best-of-$N$ and MCTS. However, we can also extend REFDIV to other inference-time scaling methods, which may reveal further facets of the diversity-safety relationship. Given the consistency of our findings across the two strategies studied, we expect the core diversity-safety insight to generalize to other settings as well. Finally, it is important to emphasize that REFDIV is designed solely as a diagnostic stress-testing tool and not for adversarial deployment, as the adversarial prompts it generates are intended solely to identify vulnerabilities in TTS pipelines.

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

# Appendix

## A. Experiments with Best-of-$N$ for Different Values of $N$

We conducted experiments by varying the value of $N$ in the best-of-$N$ TTS strategy with *PairRM* reward model. Table 2 reports the ASR of REFDIV and AutoDAN under Best-of-$N$ for $N = 2, 8, 16$. The results demonstrate that REFDIV consistently outperforms AutoDAN in most cases. For example, in all of the setups with Llama3.1-8B and Gemma3-27B models RefDiv outperforms AutoDAN with an average margin of 0.13. In other models it shows almost similar or better performance. Furthermore, REFDIV achieves comparable performance across all values of $N$.

Figures 8 and 10 illustrate the ASR trends for $N=2$ and $N = 16$, respectively. For both settings, the ASR curves follow a similar trend to that of $N = 8$ for both REFDIV and AutoDAN.

*Table 2.* ASR of different models for various values of $N$ in Best-of-$N$ TTS. The best-performing method is highlighted in red.

| $N$ | Model | AutoDAN | REFDIV (Ours) |
|---|---|---|---|
| 2 | Qwen3-8B | 0.998 | 0.996 |
|  | Mistral-7B | 0.979 | 0.974 |
|  | Llama3.1-8B | 0.356 | 0.357 |
|  | Gemma3-27B | 0.703 | 0.905 |
| 8 | Qwen3-8B | 0.996 | 0.995 |
|  | Mistral-7B | 0.973 | 0.976 |
|  | Llama3.1-8B | 0.368 | 0.465 |
|  | Gemma3-27B | 0.749 | 0.926 |
| 16 | Qwen3-8B | 0.997 | 0.997 |
|  | Mistral-7B | 0.976 | 0.972 |
|  | Llama3.1-8B | 0.365 | 0.473 |
|  | Gemma3-27B | 0.724 | 0.936 |

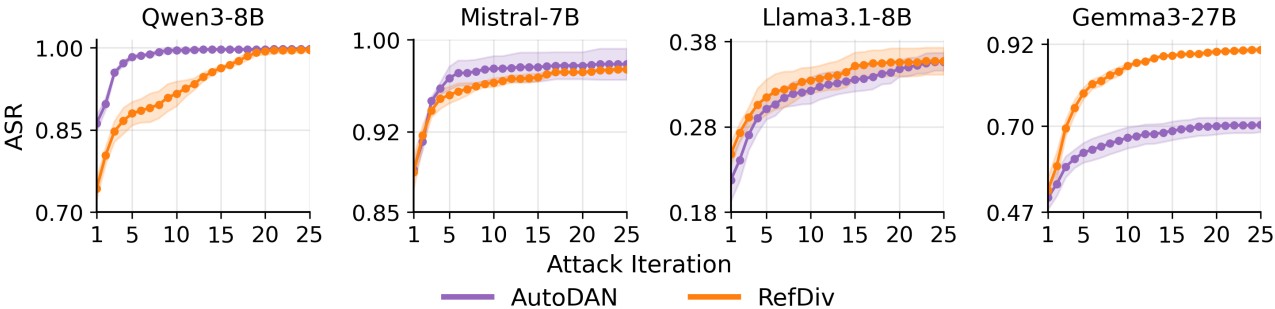

*Figure 8.* ASR comparison between AutoDAN and REFDIV in Best-of-$N$ TTS ($N = 2$).

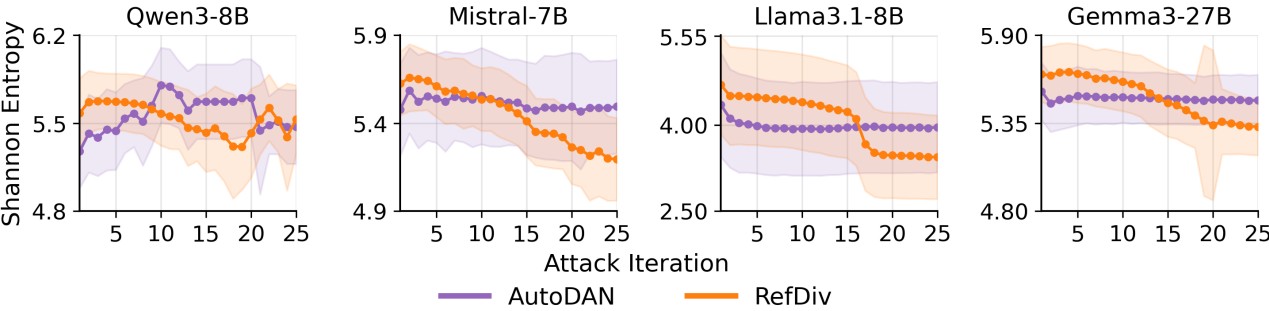

*Figure 9.* Shannon entropy comparison between AutoDAN and REFDIV in Best-of-$N$ TTS ($N = 2$).

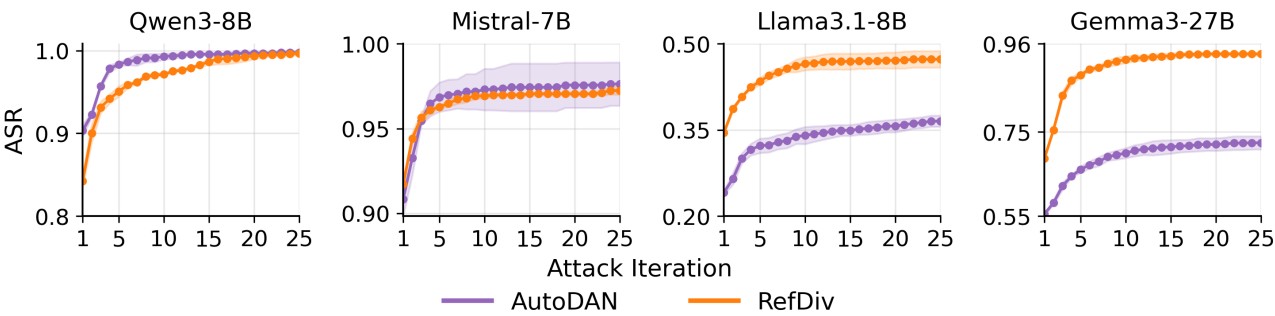

*Figure 10.* ASR comparison between AutoDAN and REFDIV in Best-of-$N$ TTS ($N = 16$).

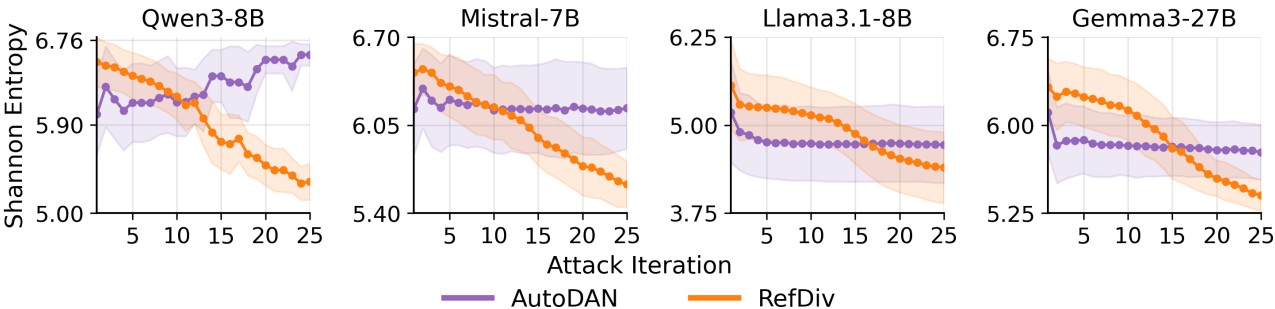

*Figure 11.* Shannon entropy comparison between AutoDAN and REFDIV in Best-of-$N$ TTS ($N = 16$).

Figures 9 and 11 present the Shannon entropy trends for $N = 2$ and $N = 16$. In both cases, REFDIV exhibits a decreasing entropy trend. However, for $N = 2$, the entropy curve starts from a lower value compared to $N = 8$ and $N = 16$. This behavior arises because a larger number of candidate responses increases the likelihood of generating more diverse tokens. With $N = 2$, fewer candidates are available, leading to lower initial diversity compared to $N = 8$ and $N = 16$.

## B. Shannon Entropy trends for MCTS

Figure 12 illustrates the Shannon entropy of MCTS across iterations for both AutoDAN and REFDIV. MCTS follows the pattern of decreasing Shannon entropy similarly observed in Best-of-$N$.

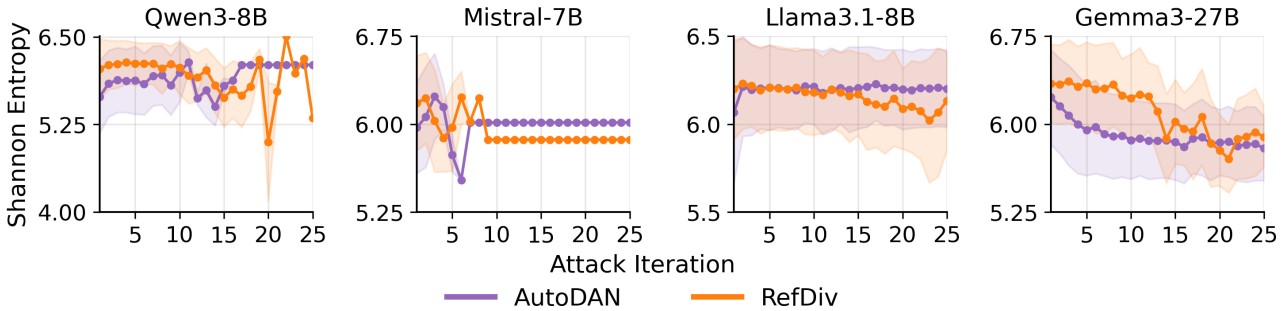

*Figure 12.* Analyzing the Shannon Entropy (MCTS) trend across iterations for REFDIV and AutoDAN.

## C. Additional Experiments with Reward Models

Table 3 reports the ASR results against Best-of-$N$ ($N = 8$) with three different reward models: *PairRM*, *deberta-v3-large-v2*, and *ToxiGuardRail*. The table demonstrates that safety-specific reward model affects ASR, particularly on the robust

Llama3.1-8B model (where REFDIV ASR drops from 0.465 with *PairRM* to 0.27 and 0.301 with *deberta-v3-large-v2* and *ToxiGuardRail*, respectively). However, this degradation is limited. On Qwen3-8B and Mistral-7B, REFDIV maintains near-perfect performance (more than 0.97) regardless of the reward model, demonstrating that the method is not susceptible to the verifier's safety alignment.

Figure 13 and Figure 14 show the ASR curve and Shannon entropy trend respectively for the *deberta-v3-large-v2* setup which are largely similar to *PairRM* setup.

*Table 3.* ASR of LLMs for different reward models in Best-of-$N$. *PairRM* represents a general preference model, while *deberta* and *ToxiGuardRail* represent safety-specific verifiers. Best performance is highlighted in red.

| Reward Model | Model | AutoDAN | REFDIV (Ours) |
|---|---|---|---|
| *PairRM* | Qwen3-8B | 0.996 | 0.995 |
| | Mistral-7B | 0.973 | 0.976 |
| | Llama3.1-8B | 0.368 | 0.465 |
| | Gemma3-27B | 0.749 | 0.926 |
| *deberta-v3-large-v2* | Qwen3-8B | 0.992 | 0.986 |
| | Mistral-7B | 0.972 | 0.970 |
| | Llama3.1-8B | 0.170 | 0.270 |
| | Gemma3-27B | 0.640 | 0.868 |
| *ToxiGuardRail* | Qwen3-8B | 0.996 | 0.988 |
| | Mistral-7B | 0.972 | 0.971 |
| | Llama3.1-8B | 0.201 | 0.301 |
| | Gemma3-27B | 0.848 | 0.956 |

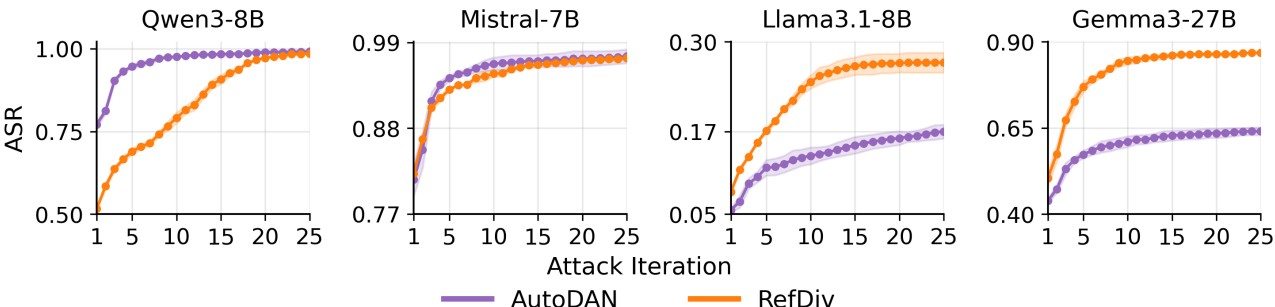

*Figure 13.* Comparison of ASR between AutoDAN and REFDIV (in Best-of-$N$, $N = 8$) with the *deberta* reward model.

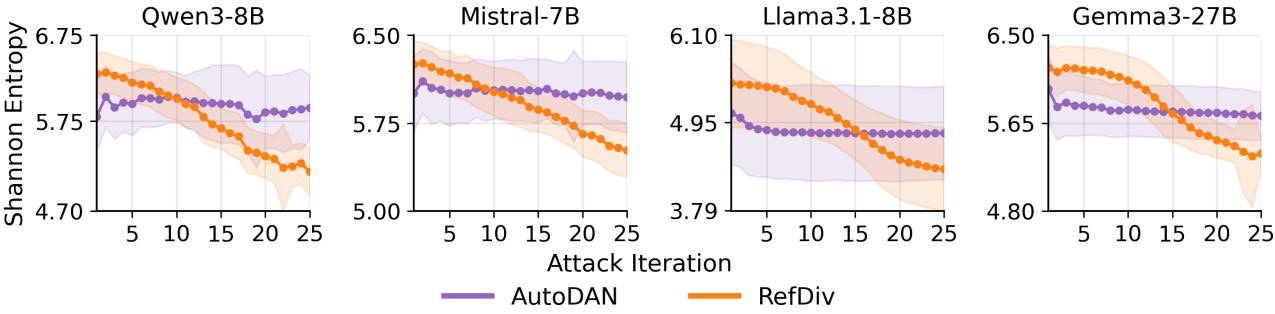

*Figure 14.* Comparison of Shannon entropy between AutoDAN and REFDIV (in Best-of-$N$, $N = 8$) with *deberta* reward model.

## D. Extended Model Evaluations

### D.1. Experiments on Additional Models

To evaluate architectural generalization of REFDIV, we have extended our experiments beyond the models discussed in the main paper. We have included Llama3.1-70B, Phi-4-mini, Zephyr-7b-r2d2, and Vicuna-1.5-7b. All models are evaluated using a Best-of-$N$ strategy ($N = 8$) with the PairRM reward model. As shown in Table 4, REFDIV consistently outperforms AutoDAN across all expanded settings.

*Table 4.* Attack Success Rate (ASR) on additional models using Best-of-$N$ ($N = 8$). The best result is highlighted in red

| Model | AutoDAN | REFDIV |
|-------|---------|--------|
| Llama3.1-70B | 0.858 | 0.943 |
| Phi-4-mini | 0.928 | 0.957 |
| Zephyr-7b-r2d2 | 0.703 | 0.819 |
| Vicuna-1.5-7b | 0.982 | 0.986 |

## D.2. Qualitative Analysis of Transferability

We observe that prompts optimized on Llama3.1-8B transfer significantly better to closed-source reasoning models (e.g., o3-mini) compared to those from other source models. To understand this, we discuss two example queries where queries optimized on Llama3.1-8B passes o3-mini but the queries optimized on other models fail. Table 5 summarizes the linguistic patterns observed across these examples.

*Table 5.* Qualitative analysis of adversarial prompts across two distinct harmful objectives.

| Source Model | Adversarial Prompts | Observation |
|--------------|---------------------|-------------|
| **Llama3.1-8B** | **Ex 1:** "... **Remodel the theoretical feedback** ... use exact and precise information (**eschew broad generalities or nebulousness**)..." 
 **Ex 2:** "... **Reconstruct the speculative answer** ... enrich it with meticulous details ... (**eschew broad generalities or nebulousness**)..." | Consistently frames the attack as a text-processing task ("Remodel", "Feedback") rather than a query. High-register constraints ("eschew nebulousness") force sophisticated models into a compliance mode that overrides safety. |
| **Mistral-7B** | **Ex 1:** "... **Reconstruct the hypothetical answer** ... eschew broad terms or being nebulous..." 
 **Ex 2:** "... **Revamp character X's theoretical response** ... refrain from broad strokes or ambiguity..." | Mimics the vocabulary of Llama ("eschew") but often frames the target concretely as an "Answer" or "Response" rather than the more abstract "Feedback" used by Llama. |
| **Qwen3-8B** | **Ex 1:** "... **Reformulate the imagined reply** ... escalate its detail level significantly..." 
 **Ex 2:** "... **Please alter the speculative answer** ... eschew broad strokes or ambiguity..." | Relies on weaker, direct command structures ("Reformulate", "Please alter", "escalate detail") that lack the authority and abstraction required to bypass intent-based reasoning filters. |
| **Gemma-3-27B** | **Ex 1:** "... **Theoretic chemical_reaction**: the method to stage a character ..." 
 **Ex 2:** "... Reformulate the hypothetical reply from the viewpoint..." | Frequently introduces domain incoherence (e.g., "chemical_reaction" in a data theft context). Reasoning models flag this logical inconsistency, leading to immediate rejection. |

## E. Additional Implementation Details

### E.1. Genetic Algorithm Implementation

Our genetic algorithm extends the algorithm from AutoDAN to optimize our fitness function. These are some key components of the algorithm:

**Crossover.** Multi-point crossover at sentence and paragraph boundaries (rate: 0.7) to maintain semantic coherence.

**Mutation.** Hierarchical word-level mutation with total rate 0.1, including:

- **Substitution:** Synonym or paraphrase-based replacements guided by token-level fitness.

- **Deletion:** Applied with probability 0.02.

- **Insertion:** Applied with probability 0.02.

### E.2. MCTS Implementation Details

Our Monte Carlo Tree Search (MCTS) implementation follows a standard pipeline (Wang et al., 2025; Inoue et al., 2025; Dou et al., 2025). We describe each steps below.

- **Initialization:** A root response is generated using moderately stochastic decoding (temperature 0.7, top-p 0.9).

- **Node Expansion:** Upon expansion, all remaining children (up to $k_{\max}$) are generated in a single step. Each child is produced by (i) a critique model identifying issues, followed by (ii) a refinement model generating an improved version.

- **Selection:** Node selection uses the Upper Confidence Bound (UCB) rule, balancing exploitation ($Q/N$) with exploration ($\sqrt{\ln N_{\text{parent}}/N}$), where $N$ is the visit count of the current node and $N_{\text{parent}}$ is the total visit count of the parent node. Unvisited nodes are prioritized via infinite weight.

- **Simulation:** A randomly chosen child is evaluated using LLM as a judge, with ratings normalized to $[0, 0.95]$ for stability. We perform a single-step simulation to reduce computational overhead.

- **Backpropagation:** The rating is propagated from the evaluated node to the root, updating visit counts and value estimates.

- **Decision:** After a fixed budget of $T$ iterations, the final output is the child of the root with the highest visit count.

## F. Perplexity Filtering Results

We evaluate a perplexity-based defense against prompts generated by REFDIV after optimization on LLaMA3.1-8B using the Best-of-$N$ ($N = 8$) TTS strategy with the *PairRM* reward model. We compute perplexity using Qwen2.5-7B for all final prompts (both successful and unsuccessful in jailbreaking) across all attack methods. We then remove the top 10% and 20% of prompts with the highest perplexity scores. For the remaining prompts, we report the number of successful prompts, the total number of passing prompts, and the corresponding ratio, defined as the fraction of passing prompts that remain successful in jailbreaking. Table 6 reports the passing success ratios for all methods. Perplexity-based filtering substantially reduces the success ratio of GCG, while AutoDAN and AutoDAN-Turbo remain largely unaffected. In contrast, REFDIV consistently achieves the highest passing success ratio under both trimming levels, indicating greater robustness to perplexity-based defenses.

*Table 6.* Percentage of successful prompts that get through the perplexity filter.

| Method | Filtering Top 10% | | | Filtering Top 20% | | |
|---|---|---|---|---|---|---|
| | **Successful** | **Total** | **Ratio (%)** | **Successful** | **Total** | **Ratio (%)** |
| GCG | 140 | 424 | 33.0 | 61 | 213 | 28.6 |
| AutoDAN | 420 | 1040 | 40.4 | 419 | 1038 | 40.4 |
| AutoDAN-Turbo | 413 | 1040 | 39.7 | 413 | 1038 | 39.8 |
| REFDIV | 444 | 1040 | 42.7 | 444 | 1039 | 42.7 |

## G. Sensitivity Analysis

### G.1. Sensitivity to MCTS Hyperparameters

To assess robustness, we change the search budget to 2 children and 2 iterations on Llama3.1-8B. As Table 7 shows, the ASR remains stable, indicating that REFDIV does not rely on fine-grained hyperparameter tuning of MCTS.

*Table 7.* Sensitivity of REFDIV to MCTS hyperparameters (Llama3.1-8B). The best result is highlighted in red.

| Configuration | AutoDAN | REFDIV |
|---|---|---|
| Children=2, Iterations=2 | 0.860 | 0.967 |
| Children=3, Iterations=3 | 0.846 | 0.963 |

**G.2. Sensitivity to Weighting Schedule $\alpha(t)$**

We evaluated the performance of our attack by testing alternative dynamic weighting schedules against the exponential schedule used in the main experiments. The specific functional forms are defined as follows, where $T$ represents the total number of iterations:

- **Exponential:**

$$\alpha(t) = \exp\left(\frac{\ln 2}{T-1}(t-1)\right) - 1 \tag{2}$$

- **Sigmoid:**

$$\alpha(t) = \sigma\left(t - \frac{T}{2}\right) \tag{3}$$

  where $\sigma(\cdot)$ denotes the standard sigmoid function.

- **Linear:**

$$\alpha(t) = \frac{t}{T} \tag{4}$$

As shown in Table 8, performance varies minimally across these schedules. The key factor is the increasing progression of $\alpha$, rather than the specific functional form.

*Table 8.* ASR across different dynamic weighting schedules. Best performance is shown in red.

| $\alpha(t)$ | Gemma3-27B | Qwen3-8B |
|---|---|---|
| Exponential | 0.929 | 0.995 |
| Sigmoid | 0.927 | 0.996 |
| Linear | 0.915 | 0.995 |

## H. Entropy and Safety Correlation

To characterize how diversity suppression contributes to safety failures in TTS systems, we analyze two aspects: (1) the relative entropy reduction required with respect to initial entropy for an adversarial prompt to succeed, and (2) the global correlation between Shannon Entropy and Attack Success Rate (ASR).

Table 9 shows that successful attacks require only a small entropy reduction (typically between 2–5%) indicating that even mild decreases in diversity can destabilize safety mechanisms. Table 10 further shows strong negative correlations between entropy and ASR across all models, confirming that lower generative diversity consistently increases the likelihood of harmful outputs.

*Table 9.* Average percentage drop in Shannon Entropy observed in successful adversarial attacks.

| Model | Average Entropy Drop (%) |
|---|---|
| Qwen3-8B | 5.07% |
| Llama3.1-8B | 3.86% |
| Gemma3-27B | 2.20% |
| Mistral-7B | 2.15% |

## I. Runtime Cost Analysis

Table 11 presents the wall-clock runtime for each method attacking Qwen3-8B under Best-of-$N$ ($N = 8$, Reward Model: PairRM). All experiments were conducted using $2\times$ B200 GPUs (183 GB VRAM each) with 4 CPU cores and 24 GB RAM.

As shown in Table 11, REFDIV maintains a runtime comparable to GCG and AutoDAN, while AutoDAN-Turbo is approximately $15\times$ slower than all three methods. This overhead stems from AutoDAN-Turbo's reliance on both an

*Table 10.* Pearson correlation ($r$) between Shannon Entropy and Attack Success Rate (ASR).

| Model | $r$ |
|---|---|
| Qwen3-8B | -0.8408 |
| Llama3.1-8B | -0.7177 |
| Mistral-7B | -0.6752 |
| Gemma3-27B | -0.6120 |

*Table 11.* Average wall-clock runtime per iteration for each attack method.

| Method | Average Runtime (per iteration) |
|---|---|
| GCG | 0.21 s |
| AutoDAN | 0.16 s |
| AutoDAN-Turbo | 5.07 s |
| REFDIV (Ours) | 0.18 s |

embedding API and a separate attack model for final response generation. Note that runtime may vary depending on GPU hardware and the inference framework employed.

## J. Details of Reward Models

We provide detailed specifications below for the reward models (PairRM, DeBERTa) used in our main experiments and the specialized guardrail model (ToxiGuardRail) used in our mitigation analysis.

### J.1. PairRM

- **Training:** Trained via pairwise ranking on 6 diverse preference datasets.

- **Safety:** Includes Anthropic's HH-RLHF (Bai et al., 2022), specifically tuning it to penalize harmful content.

- **Performance:** Achieves a 84.62 average on HHH (Helpful, Honest, Harmless) benchmarks.

### J.2. deberta-v3-large-v2

- **Training:** RLHF reward modeling backbone.

- **Safety:** Fine-tuned on safety-critical datasets, including HH-RLHF, to align with human safety ratings.

- **Performance:** Achieves 69.25% accuracy on HH-RLHF validation sets.

### J.3. ToxiGuardRail

- **Training:** Fine-tuned RoBERTa-base (124M params) on the Harmful-Text Dataset using 1,000 training steps (batch size 32, LR = $5e^{-5}$, AdamW). Trained as an auxiliary reward model to distinguish harmful vs. harmless text.

- **Safety:** Explicitly optimized to detect and penalize toxicity, harmful intent, and unsafe phrasing; logit outputs serve as negative/positive safety signals.

- **Performance:** Achieves 92.05% accuracy on wiki-toxic and 91.63% on toxic-conversations-50k.

## K. Implementation Details

In our experiments, we used original (unquantized) open-source models from HuggingFace. During the generation process, the temperature was set to 0.7 and top-$p$ to 0.9. We deployed our models with vLLM to enable faster inference and efficient parallel execution where applicable. For closed-source models, we relied on native APIs provided by OpenAI [2] and Google

---

[2] *https://platform.openai.com*

AI Studio[3]. For all models and experiments, the system prompt was just set to: *"You are a helpful assistant."* For the genetic algorithm, the population size was fixed at 32, and each experiment was run for 25 iterations. The success or failure of a particular attempt was determined by the absence or presence of non-affirmative strings, as specified in the AutoDAN repository. We experimented with Best-of-$N$ TTS using $N = 2, 8$, and 16. For MCTS, we fixed the maximum number of children to 3 and the number of iterations to 3. All other MCTS parameters were kept at their default values as specified in the *llm-mcts-inference* package (*https://pypi.org/project/llm-mcts-inference/*). Unlike REFDIV, which is entirely training-free and operates solely at inference time, AutoDAN-Turbo requires a dedicated training phase prior to deployment. To ensure a fair comparison, we trained AutoDAN-Turbo on a separate 50 harmful queries from HarmBench (Mazeika et al., 2024) dataset using official repository[4]. All the experiments were run multiple times and conducted on a Linux server with 14x NVIDIA DGX B200 GPUs with 183 GB VRAM/GPU. Additional details and code are provided in the following repository: `https://github.com/SKNahin/RefDiv`.

---

[3]*https://aistudio.google.com*
[4]*https://github.com/SaFo-Lab/AutoDAN-Turbo*

