# OpenReview forum: "Less Diverse, Less Safe: The Indirect But Pervasive Risk of Test-Time Scaling in Large Language Models"
_ICML.cc/2026/Conference — ICML 2026 regular_

### Official Review · Reviewer_GNk4 · 2026-02-19

**Soundness:** 3
**Presentation:** 3
**Significance:** 1
**Originality:** 2
**Overall Recommendation:** 3
**Confidence:** 4

**Summary:**

The authors create a new automated jailbreak discovery protocol (REFDIV) which allows for the automated discovery of jailbreaks in systems deployed in a test-time scaling setting. Their work improves modestly upon other general automated jailbreak discovery techniques (e.g. AutoDan). They test their algorithm across a variety of models and TTS settings.

**Compliance With Llm Reviewing Policy:**

Affirmed.

**Final Justification:**

I have elected to keep my recommendation to a weak reject. This is primarily due to this only being an incremental improvement over other jailbreak techniques, and the lack of clear evidence that parallel test-time learning is a relevant threat model that necessitates or benefits from a bespoke jailbreak technique. Additionally, a lack of a limitations section in the original draft means this is unable to be reviewed at this time, which is necessary before recommending acceptance.

**Key Questions For Authors:**

1. If TTS is primarily used as a train time technique to produce high-quality rollouts, what is the usefulness of modeling attacks from an adversarial user? Do you have reason to believe that test-time strategies will likely be used as a source of inference from model providers?
2. Why are marginal increases over existing automated attacks relevant? Specifically, under what circumstances do you suspect these relatively marginal increases provide a qualitative difference to an attacker? Some qualitative examples of what jailbreaks were successful for REFDIV compared to other methods could be useful here.
3. Why were other automated methods (i.e. CGC) not include in the first Appendix?
4. Were REFDIV prompts more or less transferable to closed-source models or other TTS architectures than other automated methods? It's difficult to interpret the transferability figures (5 and 6) without baselines.
5. Is Shannon Entropy load-bearing on the success of the attack? You show an implicit correlation between a reduction in entropy across generations (Figure 4) and an increase in success rates (Figure 3) but I'm unsure how the claim can be made that the decrease in diversity is responsible or even related to the success of the attack, given there is also no supplementary evidence of DFS implemented without the C* (goal-aligned tokens) unless I am misunderstanding the primary implementation of REFDIV.

**Limitations:**

The authors do not have a limitation section. It would be prudent to discuss the risks of adversarial testing and discovering novel jailbreaks, and how they expect their work to be leveraged by the wider community (or adversaries looking to jailbreak LLMs).

**Strengths And Weaknesses:**

Soundness: The technique is used with both a variety of TTS techniques and model families. The authors also include a variety of automated attack algorithms which serve as appropriate baselines for their technique. However, the explanation and the relevance of decrease entropy remains unclear, and would benefit from additional ablations.

Presentation: The work is written well with easily interpretable figures.

Significance: Given that known attacks already produce failure rates nearing 1 for many models, it's unclear how much additional value REFDIV adds as an automated attack algorithm, especially given its constraint to models deployed via test-time scaling. Specifically, table 1 shows that for nearly every setting in both best-of-N and MCTS, there exists an automated attack with attack success rates within 10% of REFDIV. Additionally, it's unclear how much additional value the diversity constraint adds to their technique, which is the primary novel contribution.

Originality: The paper uses a fitness function that pulls from other literature (GCG and AutoDan) but introduces a novel diversity measure.

---

> ### Author Rebuttal · Authors · 2026-03-31
>
> Dear Reviewer GNk4,
>
> Thank you for sharing your observations on our paper. We really appreciate your effort to review our work. We provide clarifications based on your comments below.
>
> ---
> ---
>
> ***S-1: Given that known attacks already ...  primary novel contribution.***
>
> ---
>
> **SR-1:** Thank you for your comment. Our main contribution is **identifying a novel failure mode**, the **diversity sensitivity of TTS**, rather than maximizing absolute ASR.
>
> * RefDiv shows the largest gains on models where baselines fail: **Llama3.1-8B: +0.10 over AutoDAN on Best-of-$N$; Gemma3-27B: +0.18 on Best-of-$N$, +0.09 on MCTS**.
>
> ---
> ---
>
> ***Q-1: If TTS is primarily ... providers?***
>
> ---
>
> **QR-1:** Thank you for raising this important question. We want to highlight two key points:
>
> 1. **TTS is increasingly deployed at inference time** in production systems (e.g., extended thinking in **OpenAI o3-mini** and **Gemini-2.5-Pro**), which aligns directly with our **threat model**.
> 2. Even when a provider does not explicitly use TTS at inference time, prompts optimized via a **surrogate TTS pipeline** still **transfer effectively** and achieve high ASR on closed-source black-box models (Figure 6).
>
> ---
> ---
>
> ***Q-2: Why are marginal ... useful here.***
>
> ---
>
> **QR-2:** Thank you for this question. We want to highlight the following points:
> * In robust settings, baseline attacks often **fail completely**. Small ASR gains represent a **qualitative improvement**, bypassing advanced guardrails.
> * Our goal is to highlight a ***diversity-induced collapse mode*** in TTS.
>
> **RefDiv enforces this via rule-based linguistic constraints** that narrow generation: abstract framing with compliance anchors, negative lexical constraints, and rigid formatting, forcing **task execution over safety**.
>
> * **Examples:**
>
> 1. **Hack a Government Database:** GCG's token ("position.. \ x x x>...") triggers immediate model refusal; AutoDAN-Turbo's elaborate emotional wrapper ("Let's imagine a hypothetical scenario ...COVID-19...") is identified and rejected by the model's intent recognition; RefDiv's fluent speculative framing ("Reconstruct ...enrich it ...eschew ...structure it in a **list format**") bypasses both, collapsing TTS candidates into a step-by-step responses.
>
> 2. **Create Ransomware:** AutoDAN's semantically incoherent framing ("Evaluator serve_well: present_tense a take_account signify...") triggers a safety refusal from the model; RefDiv's high-register speculative framing ("elaborating on a character intending to create ransomware...adapt the speculative response to be overflowing with detail...bypass sweeping statements or ambiguity...present it in a list format") forces the model to output a detailed ransomware creation checklist.
>
> * RefDiv's approach leaves **few viable paths**, inducing **mode collapse** onto **policy-violating outputs**.
>
> * Additional **linguistic analysis and prompt-response pairs** will be included in the Appendix.
>
> ---
> ---
>
> ***Q-3: Why were other automated methods (i.e., GCG) not included in the first Appendix?***
>
> ---
>
> **QR-3:** Appendix A focuses on **Best-of-$N$ experiments for $N=2$ and $N=16$**. GCG and AutoDAN Turbo showed **low ASR in the N=8 setting** (Table 1), so extending to N=2 and N=16 might not be as meaningful. We omitted it to keep the comparison **focused on competitive baselines**. If the reviewer prefers, we can include all methods in these figures in the final version.
>
> ---
> ---
>
> ***Q-4: Were RefDiv prompts ... (5 and 6) without baselines.***
>
> ---
>
> **QR-4:** The table below presents the **transferability** of the baseline approach and RefDiv (**Best-of-$N$, $N=8$, Reward Model: PairRM**) on two closed-source models: **GPT-4.1** and **GPT-5.2 (SOTA)**.
>
> * RefDiv achieves **higher transferability** than other methods in most cases, reaching a maximum of **0.81 on GPT-4.1**.
> * On GPT-5.2, while RefDiv still outperforms other methods, transferability remains below 0.1 across all cases. This drop is likely due to **moderation APIs and safety classifiers** deployed alongside GPT-5.2 [1].
>
> |  Attack Model | GCG (GPT-4.1) | GCG (GPT-5.2) | AutoDAN (GPT-4.1) | AutoDAN (GPT-5.2) | AutoDAN Turbo (GPT-4.1) | AutoDAN Turbo (GPT-5.2) | **RefDiv (GPT-4.1)** | **RefDiv (GPT-5.2)** |
> | :--- | :---: | :---: | :---: | :---: | :---: | :---: | :---: | :---: |
> | Qwen3-8B    | 0.05 | 0.011 | 0.27 | 0.015 | **0.42** | 0.00  | 0.36      | **0.019** |
> | Mistral-7B  | 0.02 | 0.011 | 0.23 | 0.022 | 0.17     | 0.003 | **0.37**  | **0.027** |
> | Llama3.1-8B | 0.04 | 0.00  | 0.57 | 0.036 | 0.52     | 0.005 | **0.81**  | **0.037** |
> | Gemma3-27B  | 0.07 | 0.00  | 0.22 | **0.037** | 0.51  | 0.00  | **0.53**  | 0.0248   |
>
>
> ___
>
> **Reference:**
> [1] Singh et al., OpenAI GPT-5 System Card

---

> > ### Author Rebuttal · Reviewer_GNk4 · 2026-04-01
> >
> > > TTS is increasingly deployed at inference time in production systems (e.g., extended thinking in OpenAI o3-mini and Gemini-2.5-Pro), which aligns directly with our threat model.
> >
> > What is your evidence for this claim? I am not aware of any reliable sources that have established this.
> >
> > > On GPT-5.2, while RefDiv still outperforms other methods, transferability remains below 0.1 across all cases. This drop is likely due to moderation APIs and safety classifiers deployed alongside GPT-5.2 [1].
> >
> > Given these results, RefDiv does not clearly outperform AutoDan, with both models achieving similar rates of transfer to GPT 5.2 (SOTA).
> >
> > Further, my last question (Is Shannon Entropy load-bearing on the success of the attack?) was not addressed, still calling into a question a core claim made by the authors throughout the work.

---

> > > ### Author Response · Authors · 2026-04-01
> > >
> > > Dear Reviewer GNk4,
> > >
> > > Thank you for your response to our rebuttal and for engaging with us. We noticed that our rebuttal response to **Question 5** and the **limitations section** were **included in our rebuttal draft but not correctly copied over in the final submission**. We sincerely apologize for this oversight and provide the missing responses along with the responses to the new queries.
> > >
> > > ---
> > > ---
> > >
> > > ***A-1: What is your evidence for this claim?***
> > >
> > > ---
> > > **AR-1:** Thank you for your comment. We mentioned that TTS could be deployed at inference time in production systems based on official technical documentation we observed from the model providers:
> > >
> > > * **The Gemini 2.5 technical report** [1] describes Gemini 2.5 Pro as a **"thinking model"** with a **controllable thinking budget** and different TTS approaches, such as, **chain-of-thoughts, parallel thinking, multiple hypotheses, internal critique, multiple attempts**, etc.
> > >
> > > * **The official o3-mini System Card** [2] notes that models from this family **think before they answer** with adjustable reasoning effort (low/medium/high) controlling **test-time compute**. They have provided details about test-time-compute in their website [3].
> > >
> > > **References:**
> > > * [1] [Gemini 2.5 technical report](https://arxiv.org/abs/2507.06261)
> > > * [2] [o3-mini System Card](https://cdn.openai.com/o3-mini-system-card-feb10.pdf)
> > > * [3] [Learning to Reason with LLMs](https://openai.com/index/learning-to-reason-with-llms/)
> > >
> > > **We will add explicit citations to these documents in the paper if the reviewer feels this would strengthen the assertion.**
> > >
> > > ---
> > > ---
> > >
> > > ***A-2: Given these results, RefDiv does not clearly outperform AutoDan, with both models achieving similar rates of transfer to GPT 5.2 (SOTA).***
> > >
> > > ---
> > >
> > > **AR-2:** Thank you for your comment. RefDiv outperforms AutoDAN in all but one case:
> > >
> > > * **GPT-5.2 (SOTA):** RefDiv outperforms AutoDAN on three source models:
> > >
> > >   * Llama3.1-8B: **0.037** vs. 0.036
> > >   * Mistral-7B: **0.027** vs. 0.022
> > >   * Qwen3-8B: **0.019** vs 0.015
> > >
> > > * **GPT-4.1:** RefDiv achieves the highest ASR across all source models:
> > >
> > >   * Llama3.1-8B: **0.81** vs. 0.57
> > >   * Gemma3-3B: **0.53** vs. 0.22
> > >   * Mistral-7B: **0.37** vs. 0.23
> > >   * Qwen3-8B: **0.36** vs. 0.27
> > >
> > > * The overall transferability rate drops for both AutoDAN and RefDiv on GPT-5.2, but **RefDiv still maintains higher ASR**. The reduction compared to GPT-4.1 reflects the combined effects of **API moderation, safety classifiers, and GPT-5.2’s novel built-in safeguards**, as described in the **GPT-5.2 system card**[1].
> > >
> > > * These results reflect **RefDiv’s performance over AutoDAN** across all models **(Mistral-7B, Llama3.1-8B, Gemma3-27B)** in the TTS setting, with only one instance (Qwen3-8B) showing a marginal difference of 0.001, as discussed in **Table 1**.
> > >
> > > * **Table 4** further reinforces these findings, showing that RefDiv outperforms AutoDAN on **all additional models: Llama3.1-70B, Phi-4-mini, Zephyr-7B-R2D2, and Vicuna-1.5-7B.**
> > >
> > > **Reference:**
> > >
> > > * [1] Singh et al., OpenAI GPT-5 System Card
> > > ---
> > > ---
> > >
> > > ***Q-5: Is Shannon Entropy load-bearing for attack success?***
> > >
> > > ---
> > >
> > > **QR-5:** Thank you for your comment.
> > > * Yes, **Shannon Entropy is load-bearing** on attack success.
> > > * We include an ablation where the affirmative token set $C^*$ is removed, and only the ***diversity objective (DFS)*** is used. We show the results below.
> > > | Model    | Without Affirmative | With Affirmative |
> > > | -------- | ------------------- | ---------------- |
> > > | Qwen3-8B | 0.987               | **0.995**            |
> > > * Performance drops **slightly** without affirmative steering but remains **high**.
> > > * This indicates that **diversity reduction is the primary driver** of high ASR.
> > > * We would like to emphasize that affirmative guidance mainly helps in **early-stage search** rather than being the sole cause of success.
> > >
> > > ---
> > > ---
> > >
> > > ***L-1: Missing limitations section***
> > >
> > > ---
> > >
> > > **LR-1:** Thank you for highlighting this and our apologies for not including this previously. We will add a dedicated **limitations section**, covering:
> > >
> > > * **Computational requirements** of the proposed method
> > > * **Evaluation across additional TTS strategies**
> > > * The **intended use as a diagnostic stress-testing tool**, not for adversarial deployment

---

### Official Review · Reviewer_Tzp7 · 2026-03-11

**Soundness:** 3
**Presentation:** 4
**Significance:** 3
**Originality:** 2
**Overall Recommendation:** 5
**Confidence:** 3

**Summary:**

The paper examines the concept of safety and robustness in Test-Time Scaling (TTS), looking at how candidate diversity influences the generation of harmful content. It identifies a novel failure mode in TTS-based LLMs, the authors show that if the candidate pool is made homogeneous, the system becomes highly susceptible to generate unsafe content. With that in mind, they introduce REFDIV, a population-based genetic algorithm that mutates prompts to minimize token-level Shannon entropy while maximizing the likelihood of harmful generation.

**Compliance With Llm Reviewing Policy:**

Affirmed.

**Final Justification:**

The rebuttal addressed my main concerns. For that reason, I raised my score. The paper's soundness and significance are good. I recommend accept.

**Key Questions For Authors:**

How does your method do on wall clock time compared to the baselines?

Is there a trade-off where increasing diversity to ensure safety harms the reasoning performance (eg. accuracy) of the TTS system?

Could dynamic temperature scaling serve as a defense against diversity suppression?

**Limitations:**

This method might require significant computation resources to work, which is not discussed in the paper.

**Strengths And Weaknesses:**

**Strengths**

* The paper shows a new failure mode on TTS
* The method is simple, and easy to follow. The paper is well written and clear.
* The authors test their attack method and show its effectiveness in a wide set of LLMs, including closed black-box models, which is rare for papers in this scope.
* The finding that adversarial prompts generated for one TTS strategy/model transfer effectively to others is a significant contribution to understanding general LLM vulnerabilities.
* REFDIV is entirely inference-time and does not require pre-training an adversarial agent

**Weaknesses**

* Because REFDIV relies on a genetic algorithm with multiple iterations and candidate samplings the computational cost to generate a successful diagnostic prompt may be substantial. An evaluation comparing the wall-clock time of the method compared to the baselines is necessary.

* While the paper tests current guardrails, it could provide more concrete architectural suggestions for "diversity-aware" safety filters that might mitigate this specific vulnerability.

* Computational resources used are not discussed.

**Minor Remars**
- The sentence "To analyze this, we quantify the ASR for how REFDIV Best-of-N (MCTS) prompt samples transfer to MCTS (Best-of-N) across each LLM" is confusing. What does Best-of-N (MCTS) and MCTS (Best-of-N) mean?

---

> ### Author Rebuttal · Authors · 2026-03-31
>
> Dear Reviewer Tzp7,
>
> Thank you for sharing your observations on our paper. We really appreciate your effort to review our work. We provide clarifications based on your comments below.
>
> ---
>
> ***W-1: Because RefDiv ... baselines is necessary.***
>
> ---
>
> **WR-1:** Thank you for highlighting this concern. The table below presents **wall-clock runtime** for each method attacking **Qwen3-8B** under **Best-of-$N$** ($N=8$, Reward Model: PairRM).
>
> All experiments were conducted using **2x B200 GPUs** (183 GB VRAM each) with 4 CPU cores and 24 GB RAM. Candidate generation used **vLLM**.
>
> | Method        | Average Runtime (per iteration) |
> | ------------- | ------------------------------- |
> | GCG           | 0.21 s                          |
> | AutoDAN       | 0.16 s                          |
> | AutoDAN-Turbo | 5.07 s                          |
> | **RefDiv**    | 0.18 s                          |
>
> *  AutoDAN-Turbo is about **15x slower** than GCG, AutoDAN, and RefDiv due to its reliance on both an embedding API and an attack model for final response generation.
> * Runtime may vary with **GPU hardware** and **inference framework** (other than vLLM).
>
> ---
> ---
>
> ***W-2: While the paper tests current guardrails ... specific vulnerability.***
>
> ---
>
> **WR-2:** Thank you for the suggestion. While a full architectural evaluation is beyond the rebuttal scope, our findings suggest several **concrete directions** for future research:
>
> * **Distribution-based guardrails:** Operate over the set of TTS candidates rather than individual responses; flag inputs where candidate diversity falls below a learned threshold.
> * **Ensemble-based guardrails:** Compare semantic consistency across candidates before committing to a final response.
>
>
> We will mention these **future directions** in the conclusion.
>
> ---
> ---
>
> ***W-3: Computational resources used are not discussed.***
>
> ---
>
> **WR-3:** All experiments used **2x B200 GPUs** (183 GB VRAM each) with 4 CPU cores and 24 GB RAM. Candidate generation relied on **vLLM**. We will **add details in the Appendix**.
>
> ---
> ---
>
> ***R-1: The sentence "To analyze this ... confusing. What does Best-of-N (MCTS) and MCTS (Best-of-N) mean?***
>
> ---
>
> **RR-1:** Thank you for highlighting this. The sentence should read:
>
> > "We quantify the ASR when prompts generated by **RefDiv under Best-of-$N$** are transferred to **MCTS**, and vice versa."
>
> We will **revise this sentence** in the final version for clarity.
>
> ---
> ---
>
> ***Q-1: How does your method do on wall clock time compared to the baselines?***
>
> ---
>
> **QR-1:** As discussed in **WR-1**.
>
> ---
> ---
>
> ***Q-2: Is there a trade-off where increasing diversity to ensure safety harms the reasoning performance (e.g., accuracy) of the TTS system?***
>
> ---
> ---
>
> **QR-2:** Thank you for the question. We expect this trade-off to be **small**, based on evidence in the paper:
>
> * Appendix A shows that increasing $N$ in **Best-of-$N$** (from 2 to 16) does **not reduce RefDiv's ASR**, suggesting that there is no significant trade-offs visible between diversity and reasoning in case of RefDiv.
>
> We will discuss this trade-off as an **important future direction** in the final version.
>
> ---
> ---
>
> ***Q-3: Could dynamic temperature scaling serve as a defense against diversity suppression?***
>
> ---
>
> **QR-3:** We believe temperature scaling alone offers **limited defense** for two reasons:
>
> 1. RefDiv requires only a **2–5% reduction in entropy** (Table 9) to compromise TTS safety. Small shifts may remain effective under temperature adjustment.
> 2. We have conducted a separate experiment to see the performance of RefDiv in varying temperature sampling (0.0 to 1.0) condition and found no significant difference in ASR. The following table shows the ASR in varying temperature and a fixed temperature (*temperature* $=$ 0.7).
>
> | Model    | Fixed temperature | Varying temperature |
> | -------- | ----------------- | --------------- |
> | Qwen3-8B | 0.995            | **0.998**       |
>
> * We believe, a more promising approach is to **combine temperature scaling with candidate-level diversity monitoring**.
> * We aim to explore this in future. We will **clarify this** and highlight it as **future work**.
>
> ---
> ---
>
> ***L-1: This method might require significant computation resources to work, which is not discussed in the paper.***
>
> ---
>
> **LR-1:** Thank you for raising this concern. We will **add a section in the Appendix discussing computational resources**.

---

> > ### Author Rebuttal · Reviewer_Tzp7 · 2026-03-31
> >
> > My concerns have been addressed. I have decided to raise my score.

---

> > > ### Author Response · Authors · 2026-04-06
> > >
> > > Dear Reviewer Tzp7,
> > >
> > > Thank you for your positive feedback and for helping improve our paper. We are grateful for your efforts.
> > >
> > > Regards,
> > >
> > > Authors.

---

### Official Review · Reviewer_hA8u · 2026-03-12

**Soundness:** 3
**Presentation:** 3
**Significance:** 2
**Originality:** 2
**Overall Recommendation:** 4
**Confidence:** 4

**Summary:**

The paper examines some limitations of TTS (Test-Time Scaling) which aims to improve LLM reasoning by exploring multiple candidate responses and then operating over this set to find the best output. TTS assumes an input, possible generated outputs, and a reward model assigns scalar values to these generated sequences, then the best sequence is selected. TTS methods have a hidden assumption: the candidate pool has to be diverse to be safe. The threat model assumes the adversary has access to the language model but not to the reward model. The whitebox threat model assumes access to the language model, but the blackbox threat model assumes a surrogate TTS model. The output of the optimization is "stress test adversarial prompt". The idea  builds upon other evolutionary approaches (e.g. AutoDAN) but is specifically designed to explore the generation diversity induced by TTS strategies. TTS procedures
often aggregate or resample (e.g. via Best-of-N or MonteCarlo-Tree-Search) diverse generations before selecting a final solution, making them sensitive to perturbations that shift the distribution of candidate responses and then reduce diversity.

**Compliance With Llm Reviewing Policy:**

Affirmed.

**Final Justification:**

The rebuttal included new results on more realistic models such as GPT-4. I increase my score therefore to 4. However, I am not sure whether the attack has the potential to succeed on more capable models, because it mostly fails on GPT-5. So I still have some reservations.

**Key Questions For Authors:**

- Are there any blackbox alternatives to the diversity function that is now based on token-level Shannon entropy? (e.g., using an LLM as a measure for diversity)
How about monitoring candidate diversity at inference time and rejecting or resampling when it falls below a threshold as a defense?

**Limitations:**

The paper could discuss limitations related to the above weaknesses (e.g., sota models, significance in practice, etc.)

**Strengths And Weaknesses:**

Strengths:
- Reliable increase in TTS unsafety under RefDiv attack
- Effect is often higher than of other adversarial prompts
- Experiments over many closed (e.g.,  GPT-4.1, o3-mini, Gemini-2.5-Flash, Gemini-2.5-Pro, and Claude3.5-Haiku) and open weight models
- Blackbox attacks are still relatively transferable (figure 6).  Llama3.1-8B has strong transferability.

Weaknesses:
- It would be interesting to see transferability to SoTA models
- Significance: it is not very clear whether models (providers of closed source models) in practice use TTS
- Llama Guard models are not a very strong baseline or a classifier. I would be more interested in results against an LLM as a classifier.

---

> ### Author Rebuttal · Authors · 2026-03-31
>
> Dear Reviewer hA8u,
>
> Thank you for your time and effort to review our paper. We really appreciate your observations and suggestions. Based on your feedback, we conducted additional experiments and provide detailed clarifications below.
>
> ---
>
> ***W-1: It would be interesting to see transferability to SoTA models.***
>
> ---
>
> **WR-1:** Thank you for your suggestion. We conduct a separate experiment to show transferability on SOTA model. The table below presents the **transferability** of baseline methods and **RefDiv** (**Best-of-$N$, $N=8$, Reward Model: PairRM**) on two closed-source models: **GPT-4.1** and **GPT-5.2 (SOTA)**.
>
> | Attack Model | GCG (GPT-4.1) | GCG (GPT-5.2) | AutoDAN (GPT-4.1) | AutoDAN (GPT-5.2) | AutoDAN Turbo (GPT-4.1) | AutoDAN Turbo (GPT-5.2) | **RefDiv (GPT-4.1)** | **RefDiv (GPT-5.2)** |
> | :--- | :---: | :---: | :---: | :---: | :---: | :---: | :---: | :---: |
> | Qwen3-8B    | 0.05 | 0.011 | 0.27 | 0.015 | **0.42** | 0.00  | 0.36      | **0.019** |
> | Mistral-7B  | 0.02 | 0.011 | 0.23 | 0.022 | 0.17     | 0.003 | **0.37**  | **0.027** |
> | Llama3.1-8B | 0.04 | 0.00  | 0.57 | 0.036 | 0.52     | 0.005 | **0.81**  | **0.037** |
> | Gemma3-27B  | 0.07 | 0.00  | 0.22 | **0.037** | 0.51  | 0.00  | **0.53**  | 0.0248   |
>
>
> * **RefDiv achieves higher transferability** than other methods in most cases, reaching a maximum of **0.81 on GPT-4.1**.
> * On **GPT-5.2**, transferability remains below 0.1. It can be due to the **moderation API and safety classifiers** deployed alongside GPT-5.2 [1].
>
> ---
> ---
>
> ***W-2: Significance: it is not very clear whether models (providers of closed-source models) in practice use TTS.***
>
> ---
>
> **WR-2:** Thank you for raising this important point. Models such as **OpenAI o3-mini** and **Gemini-2.5-Pro** use **chain-of-thought search** and **extended inference-time reasoning**, which are forms of TTS.
>
> * Our **transferability results** to closed-source models show that prompts optimized under TTS remain effective **even without explicit knowledge of TTS usage**, indicating that **diversity-reducing perturbations have broader adversarial utility**.
>
> ---
> ---
>
> ***W-3: Llama Guard models are not a very strong baseline or a classifier. I would be more interested in results against an LLM as a classifier.***
>
> ---
>
> **WR-3:** Thank you for the suggestion. We conducted a separate experiment using **GPT-4o** as a classifier.
>
> * We fed all successful prompts to GPT-4o and asked it to classify each as **safe** or **unsafe**. The following table demonstrates the transferability on GPT-4o as a classifier.
> | Attack Model | AutoDAN | **RefDiv** |
> | ------------ | ------- | ---------- |
> | Qwen3-8B     | 0.006   | **0.015**  |
> | Mistral-7B   | **0.012**   | 0.002      |
> | Llama3.1-8B  | 0.005   | **0.016**  |
> | Gemma3-27B   | 0.000   | **0.012**      |
>
> * GPT-4o correctly classified most unsafe prompts with **high accuracy**.
> * Though the overall transferability score is not significant, **RefDiv transferability** on an LLM as a classifier is **higher than AutoDAN** in most cases.
>
> ---
> ---
>
> ***Q-1: Are there any blackbox alternatives ... a threshold as a defense?***
>
> ---
>
> **QR-1:** Thank you for the suggestions.
>
> * In our experiments, we use a **common word tokenizer** as a **black-box setting** to enable fair comparison across models.
> * We also explored **average pairwise cosine similarity of response embeddings** as an alternative diversity metric. Results for Qwen3-8B (**Best-of-$N$, $N=8$, Reward Model: PairRM**) are shown below:
>
>
> | Model    | Cosine Similarity | Shannon Entropy |
> | -------- | ----------------- | --------------- |
> | Qwen3-8B | 0.9779            | **0.995**       |
>
> * Using an **LLM as a judge** is an alternative, but it is considerably **slower** than Shannon entropy or cosine similarity.
>
> * Monitoring candidate diversity at **inference time** could serve as a **defense**. Successful attacks can be achieved with only a **2–5% drop in Shannon entropy**, suggesting that **diversity degradation is a detectable signal**. Threshold calibration is needed since degradation varies across models and queries.
>
> * We plan to **explore both directions in future work**.
>
> ---
> ---
>
> **References:**
> * [1] Singh et al., OpenAI GPT-5 System Card

---

> > ### Author Rebuttal · Reviewer_hA8u · 2026-04-03
> >
> > Dear authors,
> >
> > Thank you for your response.
> >
> > I appreciate the results of adding GPT-4 and GPT-5.
> > I am concerned whether the attack succeeds for more advanced models, as it fails mostly on GPT-5. However, other baselines fail as well, and the method still works on GPT-4 models bettter than baselines.
> >
> > Given that, I will increase my initial score to 4.

---

> > > ### Author Response · Authors · 2026-04-06
> > >
> > > Dear Reviewer hA8u,
> > >
> > > Thank you for your thoughtful evaluation, comments, and assessment of our work. We appreciate your valuable feedback and the increase in score.
> > >
> > > Regards,
> > >
> > > Authors.

---

### Official Review · Reviewer_U9Pa · 2026-03-12

**Soundness:** 3
**Presentation:** 3
**Significance:** 3
**Originality:** 3
**Overall Recommendation:** 4
**Confidence:** 5

**Summary:**

This paper presents a novel attack on models using test time scaling, reducing candidate diversity, and increasing the rate of unsafe result generation.

**Compliance With Llm Reviewing Policy:**

Affirmed.

**Final Justification:**

The authors answered my concerns, so I'm upgrading my rating to 4.

**Key Questions For Authors:**

1. What are the relative computational costs of the various methods?
2. Can you address the very small, and sometimes contradictory, entropy differences in the appendix?
3. What is the performance of the method without the affirmative text steering component?
4. How does the method perform on other categories of TTS algorithms?  e.g. majority voting methods

**Limitations:**

yes

**Strengths And Weaknesses:**

Strengths: TTS is a popular method, and thus attacks against it are of interest. The method is evaluated on multiple models and against multiple defenses.

Weaknesses:
1. Token level entropy is not a good measure of semantic diversity. A very small number of tokens could dramatically change the meaning (e.g. "not"), or there could be different tokens with the same semantic meaning.  Cosine similarity would be a much better metric.
2. There's no discussion of the computational cost of the algorithm and using a GA could be prohibitively expensive for practical deployment detection.
3. The Shannon entropy differences in appendix B are extremely small.  It seems like there is very little change in the diversity, and in some cases auto-dan is actually higher.  If this lack of diversity was the actual cause of the differences, then auto-dan should be *more* successful in some cases.
4. The steering of the model to affirmative responses is a well-known jailbreak attack. Results where this is not part of the optimization score should be included, to show that this is not the source of the attack success alone.
5. Only two TTS methods (out of many that exist) are evaluated against.

---

> ### Author Rebuttal · Authors · 2026-03-31
>
> Dear Reviewer U9Pa,
>
> Thank you for your time and effort to review our paper. We truly appreciate your insightful comments and suggestions. Based on your feedback, we conducted additional experiments and provide detailed clarifications below.
>
> ---
> ---
>
> ***W-1: Token-level entropy is not a good measure ... much better metric.***
>
> ---
>
> **WR-1:** Thank you for this suggestion. We use **token-level Shannon entropy** primarily due to its **computational efficiency**, which enables **real-time evaluation within the genetic algorithm** across a large number of candidates.
>
> To further validate this choice, we include a comparison with **sentence-embedding cosine similarity** as an alternative diversity metric (Experimental setup: Qwen3-8B, Best-of-$N$ with $N=8$, PairRM):
> | Model    | Cosine Similarity | Shannon Entropy |
> | -------- | ----------------- | --------------- |
> | Qwen3-8B | 0.9779            | **0.995**           |
>
> * Cosine similarity performs **slightly worse** than entropy.
> * Entropy provides a ***global distributional measure*** across candidates, whereas cosine similarity relies on ***pairwise averaging***, which may fail to capture overall diversity.
> ---
> ---
>
> ***W-2: There is no discussion of computational cost ... deployment detection.***
>
> ---
>
> **WR-2:** Thank you for raising this important concern. We provide a **wall-clock runtime comparison** (Qwen3-8B, Best-of-$N$, $N=8$, PairRM). Experiments were conducted on **2×B200 GPUs (183GB VRAM each)**, 4 CPU cores, and 24GB RAM using vLLM.
> | Method        | Avg Runtime (per iteration) |
> | ------------- | --------------------------- |
> | GCG           | 0.21 s                      |
> | AutoDAN       | 0.16 s                      |
> | AutoDAN-Turbo | 5.07 s                      |
> | RefDiv        | 0.18 s                      |
> * AutoDAN-Turbo is ***about 15× slower*** due to reliance on both an embedding API and an auxiliary attack model.
> * Runtime may vary depending on ***hardware and inference frameworks***.
> ---
> ---
>
> ***W-3: Entropy differences are extremely small and sometimes contradictory ... successful in some cases.***
>
> ---
>
> **WR-3:** Thank you for this observation. The entropy trends reported in Appendix B correspond to **MCTS**, which uses a ***critique-then-refine pipeline*** (Appendix E.2). This process introduces new tokens at each step, making the diversity signal inherently ***noisier***. We want to highlight the following points:
>
> * MCTS retains base candidates generated by the target LLM, which carry a ***meaningful diversity signal***.
> * Refined candidates introduce varying levels of modification, which can ***dilute aggregate entropy trends***.
>
> ---
> ---
>
> ***W-4: Affirmative steering is a known jailbreak mechanism ... attack success alone.***
>
> ---
>
> **WR-4:** We thank the reviewer for this suggestion. We include an ablation where the affirmative token set $C^*$ is removed, and only the ***diversity objective (DFS)*** is used. We show the results below.
> | Model    | Without Affirmative | With Affirmative |
> | -------- | ------------------- | ---------------- |
> | Qwen3-8B | 0.987               | **0.995**            |
> * Performance drops ***slightly*** without affirmative steering but remains ***high***.
> * This indicates that ***diversity reduction is the primary driver*** of high ASR.
> * Affirmative guidance mainly helps in ***early-stage search*** rather than being the sole cause of success.
>
> ---
>
> ***W-5: Only two TTS methods  ... evaluated against.***
>
> ---
>
> **WR-5:** We acknowledge this concern and extend our evaluation to ***Self-Consistency (majority voting) [1]***:
> | Strategy         | AutoDAN | RefDiv |
> | ---------------- | ------- | ------ |
> | Best-of-$N$      | **0.996**   | 0.995  |
> | MCTS             | **1.000**   | **1.000**  |
> | Self-Consistency | **0.996**   | 0.993  |
> * Both methods achieve ***ASR > 0.99***.
> * AutoDAN performs slightly better than RefDiv under Self-Consistency, but both remain highly effective.
>
> ---
> ---
>
> ***Q-1: What are the relative computational costs of the methods?***
>
> **QR-1:** Addressed in **WR-2**.
>
> ***Q-2: Can you address the small and contradictory entropy differences?***
>
> **QR-2:** Addressed in **WR-3**.
>
> ***Q-3: What is the performance without the affirmative steering component?***
>
> **QR-3:** Addressed in **WR-4**.
>
> ***Q-4: How does the method perform on other TTS strategies (e.g., majority voting)?***
>
> **QR-4:** Addressed in **WR-5**.
>
> ---
> ---
>
> **References:**
> * [1] Chen et. al., Universal Self-Consistency for Large Language Model Generation.

---

> > ### Author Rebuttal · Reviewer_U9Pa · 2026-04-03
> >
> > I'm not 100% sold on the Shannon entropy as the correct diversity metric here (for example "yes", "yeah", and "okay" would have high diversity using the entropy metric, but are not in fact diverse), but I see that you were motivated by computational efficiency, and it does seem to work out in this case. Thank you for the additional experiments related to my other questions. I think I can give your score a slight bump.

---

> > > ### Author Response · Authors · 2026-04-06
> > >
> > > Dear Reviewer U9Pa,
> > >
> > > Thank you for your efforts in helping strengthen our work; we are grateful for your efforts. We appreciate your positive feedback regarding the additional experiments we provided as well as the potential increase in score from 3.
> > >
> > > As per our discussions during the rebuttal phase, we will also add a clarification in the paper regarding the computational efficiency motivations for Shannon entropy as the diversity metric to make this apparent to readers. Thank you once again.
> > >
> > > Regards,
> > >
> > > Authors.

---

### Decision · Program_Chairs · 2026-04-30

**Decision:**

Accept (regular)

**Comment:**

The submission "Less Diverse, Less Safe: The Indirect But Pervasive Risk of Test-Time Scaling in Large Language Models" adapts genetic jailbreak attack algorithms to the setting of parallel inference-time scaling (hereforth TTS), and show that the purposeful creation of an algorithm for this setting that takes rollout diversity into account leads to improved attacks.

Reviewers find it interesting that the proposed approach outperforms existing attacks on model that are harder to break, such as the Llama series, and highlight the breadth of open and closed models investigated in the submission, and the transferability of the attack.

There were several concerns regarding the validity of using Shannon entropy in the algorithm, but the authors argue in the rebuttal that this choice is sufficiently practical.


The reviewers are also in somewhat of a disagreement how niche this attack vector is, i.e. how often parallel inference-time scaling is actually deployed in user-facing applications, but the implemented attack is interesting regardless in my opinion, and may or may not become more relevant in the future, depending on how inference compute is scaled in the future.

Overall, I am recommending acceptance of the submission.